# Tailoring of Rosuvastatin Calcium and Atenolol Bilayer Tablets for the Management of Hyperlipidemia Associated with Hypertension: A Preclinical Study

**DOI:** 10.3390/pharmaceutics14081629

**Published:** 2022-08-04

**Authors:** Mahmoud M. A. Elsayed, Moustafa O. Aboelez, Mohamed S. Mohamed, Reda A. Mahmoud, Ahmed A. El-Shenawy, Essam A. Mahmoud, Ahmed A. Al-Karmalawy, Eman Y. Santali, Sameer Alshehri, Mahmoud Elkot Mostafa Elsadek, Mohamed A. El Hamd, Abd El hakim Ramadan

**Affiliations:** 1Department of Pharmaceutics and Clinical Pharmacy, Faculty of Pharmacy, Sohag University, Sohag 82524, Egypt; 2Department of Pharmaceutical Chemistry, Faculty of Pharmacy, Sohag University, Sohag 82524, Egypt; 3Department of Pharmaceutics and Pharmaceutical Technology, Faculty of Pharmacy, Al Azhar University, Assiut 71524, Egypt; 4Department of Clinical Pathology, Faculty of Veterinary Medicine, Zagazig University, Zagazig 44511, Egypt; 5Department of Pharmaceutical Medicinal Chemistry, Faculty of Pharmacy, Horus University-Egypt, New Damietta 34518, Egypt; 6Department of Pharmaceutical Chemistry, College of Pharmacy, Taif University, P.O. Box 11099, Taif 21944, Saudi Arabia; 7Department of Pharmaceutics and Industrial Pharmacy, College of Pharmacy, Taif University, P.O. Box 11099, Taif 21944, Saudi Arabia; 8Department of Pharmaceutics and Industrial Pharmacy, Faculty of Pharmacy, Merit University, Sohag 82755, Egypt; 9Department of Pharmaceutical Sciences, College of Pharmacy, Shaqra University, Shaqra 11961, Saudi Arabia; 10Department of Pharmaceutical Analytical Chemistry, Faculty of Pharmacy, South Valley University, Qena 83523, Egypt; 11Department of Pharmaceutics, Faculty of Pharmacy, Port Said University, Port Said 42515, Egypt

**Keywords:** rosuvastatin calcium, atenolol, bilayer tablets, factorial design, hyperlipidemia, hypertension, preclinical studies

## Abstract

Hyperlipidemia is still the leading cause of heart disease in patients with hypertension. The purpose of this study is to make rosuvastatin calcium (ROS) and atenolol (AT) bilayer tablets to treat coexisting dyslipidemia and hypertension with a single product. ROS was chosen for the immediate-release layer of the constructed tablets, whereas AT was chosen for the sustained-release layer. The solid dispersion of ROS with sorbitol (1:3 *w*/*w*) was utilized in the immediate-release layer while hydroxypropyl methylcellulose (HPMC), ethylcellulose (EC), and sodium bicarbonate were incorporated into the floating sustained-release layer. The concentrations of HPMC and EC were optimized by employing 3^2^ full factorial designs to sustain AT release. The bilayer tablets were prepared by the direct compression method. The immediate-release layer revealed that 92.34 ± 2.27% of ROS was released within 60 min at a pH of 1.2. The second sustained-release layer of the bilayer tablets exhibited delayed release of AT (96.65 ± 3.36% within 12 h) under the same conditions. The release of ROS and AT from the prepared tablets was found to obey the non-Fickian diffusion and mixed models (zero-order, Higuchi and Korsmeyer–Peppas), respectively. Preclinical studies using rabbit models investigated the impact of ROS/AT tablets on lipid profiles and blood pressure. A high-fat diet was used to induce obesity in rabbits. Bilayer ROS/AT tablets had a remarkable effect on decreasing the lipid profiles, slowing weight gain, and lowering blood pressure to normal levels when compared to the control group.

## 1. Introduction

Oral administration is the most common route of drug administration due to its precise dose, patient acceptability, economical manufacturing process, and extended shelf-life [1]. Conventional tablets are usually associated with recurrent dosing per day and unpredictable drug plasma concentration due to gastrointestinal degradation and/or first-pass hepatic metabolism which leads to low bioavailability and short duration of activity [2]. Sustained-release tablets offer a steady-state drug plasma level to increase the therapeutic effectiveness of the drugs and reduce the toxicity associated with an extended period of treatment [3]. In recent years, the use of bilayer or multilayer tablets to combine two or more drugs into a single dosage form has increased [4]. Such tablets provide different drug release patterns (immediate with sustained drug release). They also introduce a way of avoiding the chemical incompatibilities that may occur between the administered drugs through their physical separation. Furthermore, the formulation of a single tablet instead of using two or three tablets of different drugs will improve patient compliance and increase therapeutic efficacy [5,6,7,8,9]. A bilayer tablet of ranitidine hydrochloride (RTH) and diclofenac sodium (DS) has been developed and showed a conventional release of RTH with a sustained release of DS for the effective treatment of musculoskeletal pain associated with minimum gastrointestinal complications [10,11,12].

High-fat food is the main cause of obesity worldwide. Obesity leads to an increased risk of atherosclerosis, diabetes, and essential hypertension. Although the relationship between obesity and hypertension is unclear, there are many factors, such as insulin resistance, that raise sympathetic activity and renin/angiotensin system activity and hence increase blood pressure that have been proposed [13].

Hyperlipidemia and hypertension are the main risk factors for coronary heart disease. Many individuals with hyperlipidemia are usually suffering from hypertension which requires concurrent drug administration [14,15,16,17]. Rosuvastatin calcium, ROS, a (3R, 5S)-7-[4-(4-fluorophenyl)-2-(N-methyl methane sulfonamide)-6-(propane-2-yl) pyrimidin-5-yl]-3,5-dihydroxyhepten-6-oic acid calcium [18] is a competitive inhibitor of HMG-CoA reductase, the rate-limiting enzyme that converts 3-hydroxy-3-methyl glutaryl coenzyme A to mevalonate, the precursor for cholesterol. ROS is used to reduce the progression of atherosclerosis and for the primary prevention of cardiovascular disease [19]. ROS is a BCS Class II drug, has a water solubility of 0.33 mg/mL, and exhibits poor solubility in the gastrointestinal fluids with extensive first-pass metabolism. Moreover, its oral bioavailability is ~20% and the elimination half-life is 20 h [20,21,22]. Atenolol (AT), a 4-[2-hydroxy-3-[(1-methyl ethyl) amino] propoxy] benzene acetamide, is a β_1_-selective adrenergic blocking agent. It is widely used in the management of hypertension alone or in combination with other antihypertensive agents. It belongs to BCS Class III, which is known for its high solubility and low permeability and so low bioavailability [23]. Unfortunately, AT undergoes extensive first-pass metabolism in the liver with an oral bioavailability of about 50% [24]. So, an increase in the residence time in the stomach may enhance drug absorption and improve its bioavailability [25].

The objective of the present work is to formulate a bilayer floating tablet containing a combination of ROS (in the immediate-release pattern) and AT (in the sustained-release pattern) using the direct compression method. The formulated tablets were evaluated as shown in (Figure 1), where sorbitol was employed at different ratios to prepare various SD formulations of ROS, and the formula with the highest dissolution parameters was chosen to be incorporated into the immediate-release layer of the planned bilayer tablets. The effects of different hydroxypropyl methylcellulose and ethylcellulose concentrations on AT release were investigated by a full factorial design, and the optimized formula was used in the sustained-release layer of the bilayer tablets [26]. Thus, the selected formula was used for preclinical in vivo studies using the rabbit model [27].

## 2. Materials and Methods

### 2.1. Materials

ROS and AT powders, Epirovastin (10 mg), and Ateno (50 mg) tablets were kindly provided by Egyptian International Pharmaceutical Industries CO. [EIPICO], Cairo, Egypt. Sorbitol, Croscarmellose sodium, crospovidone, and sodium starch glycolate were purchased from Sigma-Aldrich, Baden-Württemberg, Germany. Ethylcellulose (EC) (48.8% ethoxyl, 20.0 cP) was supplied by FLUKA Chemika, Buchs, Switzerland. Magnesium stearate, hydroxypropyl methylcellulose (HPMC) K100, and lactose monohydrate were purchased from El-Nasr Pharmaceutical Chemicals Co., Cairo, Egypt. A Milli-Q Reagent Water System was used to obtain the high-quality water used to produce the solutions (Continental Water Systems, El Paso, TX, USA). All other chemicals, reagents, and solvents were purchased from regular vendors and utilized without further purification.

### 2.2. Preparation of ROS Solid Dispersions (ROS-SDs)

ROS-SDs with sorbitol were prepared using the co-evaporation method [28]. Different ratios of ROS and sorbitol (1:1, 1:2, 1:3, and 1:4 *w*/*w* ROS: carrier) were selected for the preparation of SDs based on practical trials. The accurately weighed amounts of ROS and sorbitol were dissolved in a minimum amount of methanol. The solvent was allowed to be evaporated at room temperature under a vacuum until a constant weight was achieved. The residues were kept in a desiccator overnight at room temperature then pulverized and passed through a 60-mesh sieve.

### 2.3. Dissolution Study of the Prepared ROS-SDs

A dissolution study was carried out using a USP type II dissolution apparatus (VDS, Hanson Research Co., Massachusetts, Chatsworth, USA). A dissolution medium of HCl (900 mL, acidic pH 1.2) was kept at 37 ± 0.5 °C and stirred at 100 rpm for 10 min at room temperature. The pure ROS (10 mg) and SDs equivalent to 10 mg of ROS were dispersed in the media. Samples of 5 mL were withdrawn and filtered (Whatman filter paper Grade 41, 0.45 μm), at time intervals for 2 h. The amount of ROS was measured spectrophotometrically (UV-Visible Spectrophotometry, Shimadzu 1601, Koyoto, Japan) at 247 nm (*n* = 3) [29]. Thus, the percentage of the dissolution efficiency (%DE), and relative dissolution rate (RDR) within 60 min were utilized to estimate the dissolution performance of ROS in the SDs and/or pure ROS [30].

### 2.4. Differential Scanning Calorimetry Characterizations (DSC)

The DSC curves of the prepared ROS-SDs (1:3 *w*/*w*), sorbitol, and pure ROS were recorded using DSC (DSC60, Shimadzu, Kyoto, Japan). A quantity of 5 mg of different samples was placed in aluminum pans and sealed with pierced lids. The thermal behavior of the investigated samples was studied in temperature ranges of 25–250 °C by heating at 10 °C/min under a purge of nitrogen [31].

### 2.5. Analysis of Powder X-ray Diffraction (PXRD)

An automated X-ray diffractometer Philips PW 1710, Park Guildford, UK was used to analyze the different samples (pure rosuvastatin calcium, sucrose, and ROS-SDs, 1:3 *w*/*w*). CuKα radiation at 40 kV and 30 mA (lKα = 1.4309 Å) was used to detect diffraction peaks. At a scanning speed of 5°/min, the samples under investigation were scanned from 3 to 70 °C [32].

### 2.6. Fourier-Transform Infrared Spectroscopy (FT-IR) Characterizations

FT-IR spectrum of the prepared ROS-SDs (1:3 *w*/*w*) was investigated using an FT-IR spectrophotometer (Nicolet 6700, Waltham, MA, USA) and compared to that of the FT-IR spectra of the pure ROS and sorbitol. The investigated samples were mixed with a suitable amount of potassium bromide and compressed into disks using a hydraulic press and scanned from 4000 to 400 cm^−1^ [33].

### 2.7. Preparation of Immediate-Release Layer (IRL) of ROS (ROS-IRL)

Nine formulations of ROS-IRL were formulated as described in Table 1. The specific amounts of ROS-SDs (1:3 *w*/*w*) equivalent to 40 mg of ROS, croscarmellose (CCS), sodium starch glycolate (SSG), or crospovidone (CP) were passed through a 60-mesh sieve and mixed homogenously in a mortar for 15 min. Then the magnesium stearate (1.5 mg) and lactose monohydrate were passed through a 60-mesh sieve and added to the above mixture and mixed for 10 min. Finally, a red coloring agent (2 mg) was mixed with the total powder blends. The final mixture (150 mg) was compressed using a single punch tablet machine (Royal Artist, Mumbai, India) that was equipped with flat-faced 10 mm punches.

### 2.8. Preparation of AT Floating Sustained-Release Layer (SRL) (AT-SRL)

A 3^2^ full factorial design was used to study the effect of varying concentrations of HPMC (X_1_) and EC (X_2_) on the percentage of AT released (Y_1_) from the prepared tablets. The selected factors (the concentration of HPMC and EC) and the dependent response (percent drug released at 12 h) are illustrated in Table 2. Nine formulae of AT-SRL were prepared as shown in Table 3. The ingredients of different formulations were accurately weighed and sieved through a 40-mesh sieve. All the ingredients except magnesium stearate were mixed for 15 min. Then the powder blends were further mixed with magnesium stearate for 10 min and compressed by using a tablet punching machine (Royal artist, Mumbai, Maharashtra, India).

### 2.9. Regression Analysis

Using Statgraphics plus software, one-way ANOVA was used to investigate the response parameters for statistical significance (*p* = 0.05). (Statpoint Tech., Inc., Warrenton, VA, USA). Individual parameters were also explored using the F test and quadratic models which include [7,34,35,36,37]:*Y* = *β*_0_ + *β*_1_
*X*_1_ + *β*_2_
*X*_2_ + *β*_4_
*X*_1_^2^ + *β*_5_
*X*_2_^2^+ *β*_3_
*X*_1×2_(1)
where *Y* = the level of the measured response; *β*_0_ = is the intercept *β*_1_ to *β*_5_ = the regression coefficients. All of these variables are independent variables that were exploited to mimic the design sample space’s curvature in terms of their quadratic terms, which we call *X*_1_^2^ and *X*_2_^2^ [38,39,40]. In addition, the data were fitted to several predictor equations using a backward elimination methodology. As a function of *X*, the response parameter *Y* was represented graphically as a curvature surface using the quadratic models derived by regression analysis. Furthermore, the contour plots showed the influence of the independent factors on each of the response parameters. The optimal formulation variable settings were found using a numerical optimization method based on the desirability approach. In addition, by placing restrictions on the dependent and independent variables, improved formulations were developed [41,42,43].

### 2.10. Pre-Compression Characterization of Different Prepared Tablets

#### 2.10.1. Angle of Repose Study

The different prepared powder blends were allowed to flow through the funnel fixed to a stand at a definite height. The angle of repose was then calculated by measuring the height and radius of the heap of powder formed utilizing Equation (2).
tan *θ* = *h/r*(2)
where, *θ* = angle of repose, *h* = height of powder heap, and *r* = radius of the powder cone. The test was performed in triplicate and the obtained *θ* is correlated to the standard values [44].

#### 2.10.2. Bulk Density Study

An accurately weighed amount of the different prepared powders was carefully poured into a graduated glass cylinder, and then the volume was measured directly from the graduation marks on the cylinder as mL. Thus, the measured volume was the bulk volume and the bulk density was calculated by Equation (3).
*D**_b_* = *Wt/V*_0_(3)
where *Wt* = weight of the powder and *V*_0_ = bulk volume.

#### 2.10.3. Tapped Density Study

After measuring the bulk volume; the same measuring cylinder was tapped till no further change in the powder volume and the final powder volume was noted as (*V_f_*). The tapped density is calculated by Equation (4) [45].
*D**_t_* = *Wt/V*_f_(4)

#### 2.10.4. Carr’s Index (CI) and Hausner’s Ratio (HR) Study

These are considered important parameters used to characterize the nature of powder’s flow [46]. CI could be calculated by Equation (5).
*CI* = *D**_t_* − *D**_b_/D**_t_* × 100(5)

HR could be calculated by Equation (6):*HR* = *D_t_/D_b_*(6)

### 2.11. Post-Compression Tablet Evaluation

#### 2.11.1. Drug Content Study

Twenty tablets of each obtained patch were weighed individually and powdered. The drug content was calculated and expressed as a percentage of labelled claims according to USP specifications [47,48]. The test was carried out in triplicate and the mean values ± SD were calculated.

#### 2.11.2. Tablet Weight Variation Study

Twenty tablets of each batch were selected randomly and weighed individually using an electronic digital balance (AS 60/220. R2, Radwag, Torunska, Poland) and the total mean weight was calculated. No more than two tablets should lie outside the percent deviation and no tablet should differ by more than twice the limit according to the USP specifications [49].

#### 2.11.3. Tablet Thickness Study

Ten tablets of each batch were taken and their thicknesses were measured using the Caliper Vernier apparatus (Mitutoyo 530-116/Series 530, Kawasaki, Kyoto, Japan). The average thickness ± SD was calculated.

#### 2.11.4. Tablet Friability and Hardness Study

Twenty tablets of each batch were selected randomly and de-dusted using a fine brush then weighed (*Wt_i_*) and placed into the drum of the friability tester (Vinsyst, VFT, 300 Mm, Mumbai, India). The friabilator was adjusted to rotate at speed of 25 rpm for 4 min at room temperature. The tablets were de-dusted again after the end of rotation and re-weighed (*Wt_f_*). The friability was calculated by Equation (7) as the percent ratio of the loss in weight.
*Friability* (%) = *(Wt_i_* − *Wt_f_/Wt_i_)* × 100(7)

The friability percentage of the tablets less than 1% was considered acceptable [50].

For characterization of the fabricated tablets’ hardness, ten tablets of each batch were selected randomly and inserted into the middle of the hardness tester (Monsanto, VMT-1, Mumbai, India). and the force required to break the tablet was measured in kg/cm^2^.

#### 2.11.5. In Vitro Buoyancy Study

The in vitro buoyancy of floating AT-SRL mono tablets was carried out using a beaker containing simulated gastric fluid (pH 1.2, 200 mL) at 37 ± 0.5 °C. The time required for tablets to reach the surface of the medium is called floating lag time. The duration of time the tablets permanently floated on the surface was calculated as total floating time [51].

#### 2.11.6. In Vitro Drug Release Study

##### In Vitro Release of ROS-IRL Mono Tablets

In vitro drug release of different tablets of ROS (ROS1-ROS9) was studied using a USP type I dissolution apparatus (VDS, Hanson Research Co., Massachusetts; Chatsworth, USA) of simulated gastric fluid (pH 1.2, 900 mL) was used as the dissolution medium. Each basket was charged with one tablet then the apparatus was operated at a speed of 100 rpm at 37 ± 0.5 °C. A 5 mL quantity of the samples was withdrawn at a predetermined time interval of 120 min and replaced with the same volume of fresh SGF. The sample was filtered through a Whatman 0.45 μm membrane filter and analyzed spectrophotometrically, at λ_max_ of 247 nm. 

##### In Vitro Release of AT-SRL Mono Tablets

In vitro drug release of different tablets of AT was studied using the same conditions used for ROS-IRL mono tablets. The amount released was determined spectrophotometrically at 224 nm [52].

##### In Vitro Drug Release Kinetics Study

The mechanism of drug release from all the prepared tablet formulations was studied by fitting the release data in different kinetic models as described below:

Zero-order kinetics [53]
*Q* = *Q*_0_ + *K*_0_
*t*(8)
where, *Q* = the amount of drug dissolved at time *t*; *Q*_0_ = the initial amount of drug in the solution; and *K*_0_ = the zero-order release constant.

First-order kinetics [54]
*log Q* = *log Q*_0_ − *K*_1_
*t*/2.303(9)
where *K*_1_ is the first-order release constant.

Higuchi kinetics [55]
*Q* = *K_H_ t*^0.5^(10)
where *K_H_* = the Higuchi rate constant.

Korsmeyer–Peppas kinetics [56]
*M_t_/M_∞_* = *kt^n^*(11)
where *M_t_/M*_∞_ = the fraction of drug released at time *t*, *k* = the rate constant, and *n* = the release exponent. The *n* value is used to describe different release mechanisms during the dissolution process. A value of *n* ≤ 0.43 indicates that drug release is controlled by Fickian diffusion, whereas a value of *n* ≥ 0.85 suggests that drug release is dominated by an erosion mechanism. For values 0.43 < *n* < 0.85, the release is described as anomalous, implying that a combination of diffusion and erosion contributes to the control of drug release.

### 2.12. Bilayer Tablets (BLTs) Compression Study

Depending on the dissolution behavior, the ROS3 formula and the optimized formula of AT were selected to prepare BLTs. Firstly, AT (sustained-release layer) was introduced into the die cavity and compressed with compression force between 2–4 kg/cm^2^). Secondly, ROS (immediate-release layer) was fed into the same cavity above the AT SRL and compressed with compression force between 6–8 kg/cm^2^ to obtain the BLTs.

Secondly, ROS (immediate-release layer) was fed into the same cavity and compressed at 6–8 kg/cm^2^ until the overall desired hardness of XXX unit was obtained.

The prepared tablets were evaluated for their weight variation, drug content uniformity, friability, hardness, and floating time. The dissolution manner was investigated using the same conditions used for mono tablets.

### 2.13. Accelerated Stability Study

A stability study was performed to investigate the effect of storage conditions (temperature and relative humidity) on the drug content, average weight, hardness, and in vitro drug release pattern [57,58]. The selected BLTs were packed in sealed amber-colored bottles and charged at accelerated stability conditions of 40 ± 1 °C/75 ± 5% RH in the humidity chamber (temperature range: 20–60 °C, humidity range: 40–95% RH, accuracy: ± 2.0 °C, and ± 3.0% RH) for six months. Time intervals of sample withdrawal were 1, 2, 4, and 6 months.

### 2.14. In Vivo Assessment Study

#### 2.14.1. Experimental Animals

Thirty male clinically healthy New Zealand white rabbits weighing about 1000 ± 5.0 g were selected for the preclinical animal study. They were obtained from the central animal house of the Faculty of Veterinary Medicine, Zagazig University. The ZU-IACUC Committee of the Veterinary Faculty, Zagazig University, Egypt, gave its approval to this experiment’s methods for the care of experimental animals. Application number ZU-IACUC/3/Az/89/2021. The animals were housed in metal cages under the standard hygienic conditions, accommodating temperatures of 22–24 °C, and 12 h light/dark cycle. They were kept under observation and examination one week before the experiments to be sure that they were free from bacterial and parasitic infections. They were fed with free access to a standard diet and water.

#### 2.14.2. Experimental Design

The rabbits were divided randomly into three groups: Group A, control (*n* = 10) was fed a standard diet; Group B (*n* = 10), was given a standard diet containing 0.5% cholesterol and 3% soybean oil for 16 weeks; Group C (*n* = 10), was received dietary cholesterol for 12 weeks followed by oral treatment with BLTs (10 mg of ROS and 50 mg of AT)/kg b.wt with normal diet for 4 weeks. The tablets were set at the pharyngeal site to be swallowed immediately by the rabbits. The body weight was monitored during the experiment. The arterial blood pressure was monitored oscillometrically utilizing forelimb and hind limb cuffs. The heart rate was measured by sensing the pulse over the femoral artery in the upper inner thigh [59]. Blood samples of 5 mL were collected from different animals (*n* = 5) from each group into a heparinized tube from the ear vein at the beginning of the experiment (week 0) and 4, 8, 10, 12, 14, 15, and 16 weeks, after fasting for 12 h. The samples were immediately centrifuged at 4000 rpm for 10 min at room temperature and the plasma was collected and stored at −8 °C until assayed. Plasma cholesterol, high-density lipoproteins (HDL), and triglycerides (TG) were measured using colorimetric reactions with commercial kits (DiaSys, Waterbury, CT, USA). Low-density lipoproteins (LDL) were calculated using the following equation [60]:LDL = Total cholesterol − (HDL + Triglycerides/5)(12)

### 2.15. Statistical Analysis

To perform statistical analysis, we used the GraphPad Prism 6.0 software (Graph Pad Software, Inc., San Diego, CA, USA). One-way analysis of variance was used to compare the variables. In this study, differences were considered statistically significant when a *p*-value of 0.05 was observed.

## 3. Results and Discussion

### 3.1. Characterization of ROS-SDs

#### 3.1.1. In Vitro Dissolution Study

Figure 2 shows the dissolution profiles of ROS-SDs with sorbitol (1:1, 1:2, 1:3, and 1:4 *w*/*w* ROS: carrier). Within the first 60 min of dissolution, pure ROS showed 29.78% of drug dissolved, while 1:1, 1:2, 1:3, and 1:4 *w*/*w* ROS-SDs showed 47.91, 70.45, 84.25, and 81.4%, respectively. In addition, the pure drug offered poor dissolution properties with a dissolution efficiency of 5.45, 12.5, and 23.98% after 30, 60, and 120 min, respectively, while % DE and RDR values were increased for all prepared SDs formulae as computed in Table 4. The ROS-SDs (1:3 *w*/*w*) had the highest DE and RDR at different time intervals, except at 30 min the RDR value for SD4 was slightly greater than that for SD3. So the SD3 formula was selected for the next studies, as the further increment of the amount of sorbitol did not significantly impact the dissolution performance. However, the enhanced dissolution rate of ROS-SDs could be related to many factors such as the absence of drug aggregation, enhanced drug wettability, drug particle size reduction, and the reduction of interfacial tension between drug and dissolution medium [61].

#### 3.1.2. Differential Scanning Calorimetry Characterizations

Figure 3 shows the thermal behavior of the individual ROS, sorbitol, and the formulated ROS-SDs (1:3 *w*/*w*). The pure ROS showed a characteristic peak at 161 °C corresponding to its melting point. This is a primary indication that a pure drug is present in crystalline nature. The sorbitol showed an endotherm at 105 °C referring to its melting point.

The thermal behavior of ROS-SDs showed the disappearance of the drug melting endotherm which indicates that the crystalline nature of ROS converted into the amorphous form which enhanced its solubility with the aid of the matrix of sorbitol. Such a result was clarified that the enhanced dissolution rate of ROS-SDs as compared to the pure ROS was not only because of the wetting effect of the hydrophilic carrier but else due to the physical exchange of ROS from the crystalline to amorphous form in the matrix of sorbitol [62].

#### 3.1.3. FT-IR Spectroscopy Characterizations

Figure 4 shows the FT-IR spectroscopy of pure ROS, sorbitol, and the formulated ROS-SDs (1:3 *w*/*w*). The individual pure ROS had characteristic peaks at 1710, 1542, 1505, 1380, and 1327 cm^−1^ which belong to -C=O stretching, -C=N stretching, and -C–C- stretching in the aromatic ring, -C–F stretching in the aromatic ring, and the asymmetric stretching for the -S=O group, respectively. In addition, the peaks of sorbitol appeared at 3374, 2928, and 1082 cm^−1^, which correspond to -O–H stretching, C–H stretching, and C–O stretching, respectively. All these peaks (sulfonyl and carbonyl groups for drug and hydroxyl group for sorbitol), were well observed with an insignificant shifting in the spectrum of the ROS-SDs formula to confirm the absence of interaction between ROS and sorbitol.

#### 3.1.4. Powder X-ray Diffraction Analysis

The PXRD diffractograms that were produced showed that pure ROS exhibited characteristic peaks in the frequencies 2*θ* = 16.79, 23.14, and 34.06. These peaks indicate that pure ROS is crystalline in composition. However, in treated ROR powder (ROS-SDs, 1:3 *w*/*w*), both the height of the peaks and the number of peaks were reduced, which indicated that the treated ROR powder had a relatively low crystallinity.

These ROS-SDs (1:3 *w*/*w*) had decreased ROS crystallinity, as evidenced by the lower peak heights and the disappearance of some significant peaks in their PXRD patterns (Figure 5). In the ROS-SDs (1:3 *w*/*w*) studied, ROS was found to have transformed from crystalline to amorphous form [63,64].

### 3.2. Pre-Compression Characterization of Tablets

Table 5 shows that all formulations showed passable to excellent flow properties. The angle of repose in the ROS blend ranged from 16.59 to 29.23°, Although all the used excipients had good flow properties, the variation in the angle of repose may attribute to the concentration of superdisintegrant, increasing the concentration of superdisintegrant leading to a decrease in the angle of repose [65] as shown in Table 5. The angle of repose in the AT blend also ranged from 19.25 to 28.7°. Although all the used excipients had good flow properties, the variation in the obtained results may be due to the hygroscopic nature of the used cellulosic derivatives (HPMC and EC) [66] which may adsorb moisture from the atmosphere on their surface and form a cohesion force between the wetted particles and decrease flow properties [67]. The variation was obtained because the test of the different formulations took place on different days and the climatic conditions varied so the obtained results were varied but still in the accepted flowability range.

Carr’s index showed values between 11.72 and 19.25 and from 10.30 and 17.42 for ROS and AT, respectively. Hausner’s ratios ranged from 1.10 to 1.32 and from 1.09 to 1.24 for ROS and AT, respectively These results displayed the good flow property of the powder blend to be formulated by direct compression [68].

### 3.3. Characterization of the Prepared Mono Tablets

Table 6 shows the post-compression characterization parameters of different ROS and AT mono tablet batches. The drug content of the different mono tablets showed values (95.29–100.82%) which were acceptable according to the USP criteria. The prepared tablets’ weights ranged from 151.24 ± 0.5 to 140.72 ± 0.7 mg and from 204.93 ± 0.3 to 189.67 ± 0.5 mg, respectively for ROS and AT tablets. The percentage of weight variation of mono tablets from the average weight was found to be within 7.5% (*w*/*w*) which confirmed that the tablets had passed the USP weight variation test. The thickness of the prepared tablets ranged from 2.08 ± 0.06 to 1.81 ± 0.03 mm and from 3.11 ± 0.07 to 2.90 ± 0.1 mm for ROS and AT tablets respectively. The friability percent was less than 1% (ranging from 0.895 to 0.532% and from 0.773 to 0.540% for ROS and AT tablets respectively). The tablets’ hardness ranged from 4.60 to 3.25 and from 7.72 to 5.6 kg/cm^2^ for ROS and AT tablets, respectively. Such results indicated that the formulated tablets had regular drug contents, acceptable weight variations, and good mechanical strength. Accordingly, all the batches could be used for further studies.

### 3.4. In Vitro Buoyancy Study

Table 7 shows the total floating time (TFT) and buoyancy lag time (BLT) of floating sustained-release AT tablets. It was observed that the BLT was less than 15 min which allowed the tablets to float before gastric emptying and the TFT was greater than 12 h which extended the residence time of AT in the stomach to delay the drug release and increase the extent of its absorption [69]. The BLT was decreased by increasing the concentration of HPMC and decreasing the concentration of EC while the TFT was increased by increasing the concentration of EC and decreasing the concentration of HPMC. These results may be due to the hydrophilicity of HPMC and the hydrophobicity of EC.

The carbon dioxide (CO_2_) generated because of the reaction of sodium bicarbonate with the buffer (pH 1.2) was trapped and kept within the gel layer formed by the hydration and swelling of HPMC. The presence of CO_2_ decreased the density of the tablets and when the density became <1, the tablets became buoyant. The hydrophobic polymer could prevent the penetration of water into the tablet matrix and so the time required for floating would be longer [70]. On the other hand, the high permeability and tendency of HPMC to increase the wettability of tablets led to an increase in the amount of absorbed liquid medium to replace the air inside the floating tablets, thus reducing the TFT [71].

### 3.5. In Vitro Drug Release Studies

#### 3.5.1. In Vitro ROS Release from IRL Mono Tablets

Figure 6 shows the in vitro release studies that were performed for each mono tablet formulation and were compared with those of the marketed ROS tablets (Epirovastin, 10 mg). The % ROS released from the prepared mono tablets was significantly (*p* < 0.05) higher than that of the commercial tablets. The % drug released was increased when the concentration of the superdisintegrants was increased [72]. Tablet formula ROS3 which contained 15 mg CCS showed the fastest drug release rate (100% released after 45 min) when compared to the other tablets’ formulae containing SSG and CP superdisintegrants. Such results could be related to the higher swelling and rapid disintegration of tablets containing CCS into fine particles [73]. The CP has a high hydration capacity and high capillary activity which leads to the rapid disintegration of formulated tablets into larger aggregated particles. The difference in the particle size caused variation in the surface area offered to the dissolution medium and therefore the diversity in % released [74]. However, while tablets containing SSG disintegrated by the immediate absorption of water followed by quick swelling into small particles, this occurred more slowly because of the formation of a viscous gel layer by the SSG [75]. Based on the obtained results, ROS3 was selected for incorporation in BLTs.

#### 3.5.2. In Vitro AT Release from Floating SR Mono Tablets

Figure 7 shows the dissolution profiles of AT released from the different mono tablets (AT1–AT9) and the marketed tablets (Ateno, 50 mg). The commercial tablets showed a faster drug release rate than all the prepared AT-SR mono tablets; while, in the prepared tablets, HPMC and EC were used to retard drug release. Equation (13) showed the effect of HPMC (X_1_) and EC (X_2_) on % released of AT (Y_1_) from AT-SR mono: % released at 12 h (Y_1_) = 71.422 − 0.7933 X_1_ + 3.3044 X_2_ + 0.009 X_1_^2^ − 0.02933 X_1×2_ − 0.03644 X_2_^2^(13)

It is clear that the % drug release was positively affected by EC and negatively affected by HPMC. As explained from the main effects plot (Figure 8A), the increase in the concentration of EC from 5 to 20% and decrease in the concentration of HPMC from 40 to 20% increased the % release of the AT from 63.6% ± 2.26 to 98.7% ± 3.42. The existence of HPMC in the tablet matrix would result in the formation of a viscous gel on the tablet surface upon contact with the dissolution medium. Depending on the density of the gel layer, the drug release would be retarded. Increasing HPMC concentration leads to the formation of a denser gel layer, so, the drug takes a long time to cross it and reach the release medium [6,76,77]. Furthermore, the presence of a water-soluble drug in the matrix of HPMC creates an extra osmotic gradient which accelerates the swelling rate of HPMC and enhances the density of the gel [78,79]. Even though the incorporation of EC in the tablet matrix may control drug release, the blending of this polymer with the matrix increased the drug release rate. This might be because EC is a large hydrophobic molecule that forces a discontinuity in the gel layer formed by HPMC, which at that time reduces the barrier for drug release, as seen in similar results demonstrated by Gupta et al. [80]. The effect of each independent variable on the release of AT was statistically significant (*p* < 0.05) as displayed in the contour plot, Figure 8B. The quadratic effects of HPMC and EC were statistically insignificant (*p* < 0.05); however, the interactive effect was statistically significant (*p* < 0.05) as exhibited in Table 8. Furthermore, the relation between the independent factors (X_1_ and X_2_) and the % released was offered through the response surface plot (Figure 8C), where, each factor’s effect on the % released was notable by the low level of the other factors. The optimization was completed using multiple response optimizations. The optimum concentrations of HPMC and EC were 20.2 and 19.96% *w/w*, respectively, which were obtained by the design software. The predicted % released was 98.67%, while the observed value was 98.1% which confirmed the validity of the regression model [81].

### 3.6. In Vitro Drug Release Kinetic Modeling

The mathematical representation of the in vitro release profiles of ROS and AT from the prepared mono tablets is listed in Table 9. The highest degree of correlation coefficient (r^2^) defines the suitable mathematical model that follows drug release kinetics [82]. It was found that the first-order model showed the highest degree of r^2^ for all formulations of ROS. While, in the case of AT mono tablets, the zero-order model exhibited the highest r^2^ in the case of AT4, AT5, and AT8 formulations, and the remaining formulations were best fitted in the Higuchi-diffusion model. To evaluate the mechanism of drug release, the release data were fitted to the Korsmeyer–Peppas exponential model. The values of n (diffusional exponent) were greater than 0.45 and less than 0.85. This indicates a non-Fickian diffusion mechanism, where the diffusion and relaxation rates are similar [83]. 

### 3.7. Formulation and Evaluation of BLTs

The bilayer tablets are composed of an ROS immediate-release layer and an AT sustained-release layer. The first layer included the ROS-SDs (1:3 *w*/*w*) and croscarmellose sodium (10% *w*/*w*). However, the second layer consisted of AT in the matrix of HPMC (20.2% *w*/*w*) and EC (19.96% *w*/*w*) to obtain a total weight of 350 mg. The drug content of the BLTs was found to be 99.65 ± 1.6 and 98.54 ± 2.3% for ROS and AT respectively. The average weight was found to be 347.51 ± 4.11 mg, within the standard limit of 100 ± 5%. The hardness of the BLTs was found to be 6.43 ± 0.21 kg/cm^2^ and the % of friability was less than 1% *w*/*w*. Such results were acceptable according to the pharmacopoeia specifications [84]. Furthermore, the total floating time and buoyancy lag time were found to be 18.01 ± 1.02 h and 7.43 ± 0.75 min, respectively.

#### In Vitro Dissolution Investigation of BLTs

Figure 9 shows the in vitro dissolution profiles of ROS and AT from the fabricated BLTs. The dissolution study revealed that 92.33% of ROS was released within 1 h in the buffer (pH 1.2). This is related to the rapid disintegration of the immediate-release layer, followed by the prompt dissolution of ROS-SDs. The other sustained-release layer of BLTs exhibited the delayed release of AT in the same buffer. This was attributed to the presence of HPMC and EC, which can be used to control the release of water-soluble drugs. The % of AT released was 21.32 ± 1.56, 62.48 ± 3.35, and 96.65 ± 3.36 after 1, 6, and 12 h, respectively.

### 3.8. Stability Studies

Table 10 shows the fabricated bilayer tablets which exhibited no noticeable changes in the evaluated parameters, such as the hardness, drug contents, average weights, and release profiles after six months of storage at the accelerated stability conditions.

### 3.9. In Vivo Study

Rabbits are widely used as a model to investigate hypercholesterolemia as their lipoprotein profile is comparable to humans [26]. As shown in Table 11, the plasma levels of total cholesterol (TC), LDL, and TG were increased significantly (*p* < 0.05) in rabbits of groups B and C (*n* = 20) after 12 weeks of nutrition with the dietary cholesterol as compared with group A, which was fed with a normal diet. The HDL decreased significantly (*p* < 0.05) in groups B and C compared to group A after week 12. After administration of BLTs to group C for four weeks, the levels of TC, LDL, and TG were decreased by 54.4, 44.6, and 47.5%, respectively. Furthermore, the level of HDL was increased in group C after 12 weeks compared to the untreated group. The elevation of the lipid profile in group B until the end of the experiment could be related to the food enriched with cholesterol which resulted in increased production of TG. Such hypertriglyceridemia directly affects the components and metabolism of HDL and LDL. The LDL is precipitated in the wall of arteries and is the primary component of atherosclerotic plaque [85]. The body weight was increased in group B from the beginning of the experiment till week 16, but in the treated group till week 12, in comparison to the control group as presented in Table 12. The heart rate and blood pressure of the rabbits fed with a high cholesterol diet were higher than that of the control. while the rabbits treated with the prepared BLTs had blood pressures similar to the control group as demonstrated in Table 12. These results indicated that ROS/AT in BLTs have a particular management role in decreasing the risk for the promotion and progression of atherosclerosis and, hence, cardiovascular diseases, through reducing the lipid parameters, delaying weight gain, and reducing blood pressure.

## 4. Conclusions

New bilayer tablets (BLTs) were fabricated and evaluated preclinically for their efficiency and safety in the management of hyperlipidemia and hypertension. However, the solubility and the dissolution rate of ROS in its carrier matrix, sorbitol, were, respectively, enhanced and improved compared to the pure ROS. The floating sustained-release layer of AT was formulated with 10% *w*/*w* sodium bicarbonate, 20.2% *w*/*w* of HPMC, and 19.96% *w*/*w* of EC. The prepared BLTs had regular drug contents, acceptable weight variations, and good mechanical strengths, as the displayed results indicate. Moreover, the in vitro release studies that were performed for each mono tablet formulation in comparison to those of the marketed tablets of ROS (Epirovastin, 10 mg), and AT (Ateno, 50 mg), showed a significantly (*p* < 0.05) higher % in the case of the poorly soluble ROS, and a slower pattern in the case of AT. The in vitro dissolution profiles of ROS and AT from the fabricated BLTs showed that 92.33% of ROS was released within 1 h in the buffer (pH 1.2), and inversely AT in the sustained-release layer exhibited a delayed release in the same buffer. Furthermore, in the in vivo rabbits model (*n* = 20), those which were treated with BLTs had blood pressures similar to the control group (*n* = 20), indicating that the ROS/AT BLTs have a particular management role in decreasing the risk for the promotion and progression of atherosclerosis and, hence, cardiovascular diseases, by reducing the lipid parameters, delaying weight gain, and reducing blood pressure. Finally, all the obtained results achieved the objective of the study, and further in vivo investigations of the preclinical pharmacokinetic parameters are needed, in order to confirm the suitability of the proposed project for human applications and commercial pharmaceutical production.

## Figures and Tables

**Figure 1 pharmaceutics-14-01629-f001:**
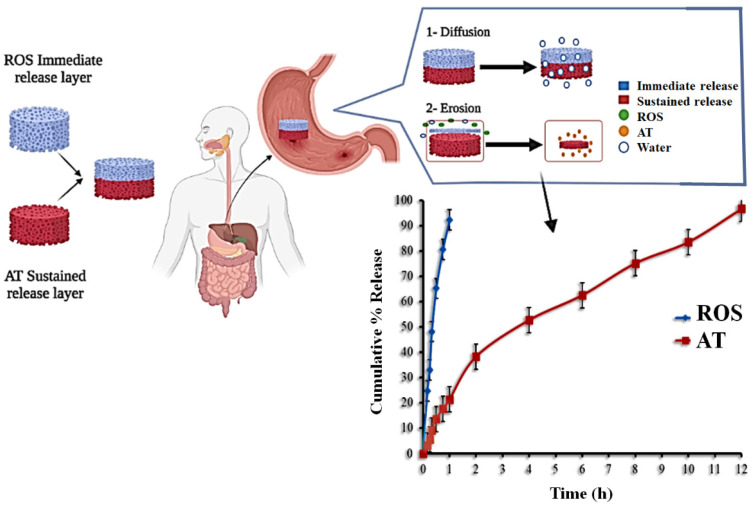
Schematic diagram of ROS/AT bilayer floating tablet design experiment.

**Figure 2 pharmaceutics-14-01629-f002:**
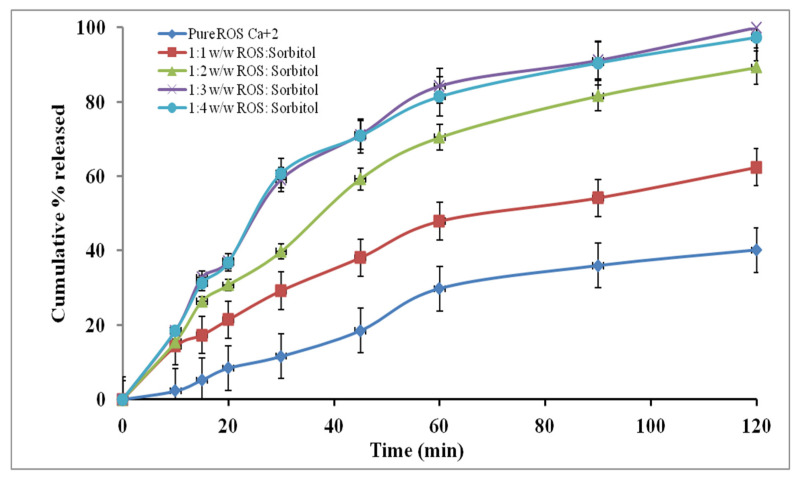
Dissolution profiles of pure ROS and ROS-SDs (*n* = 3).

**Figure 3 pharmaceutics-14-01629-f003:**
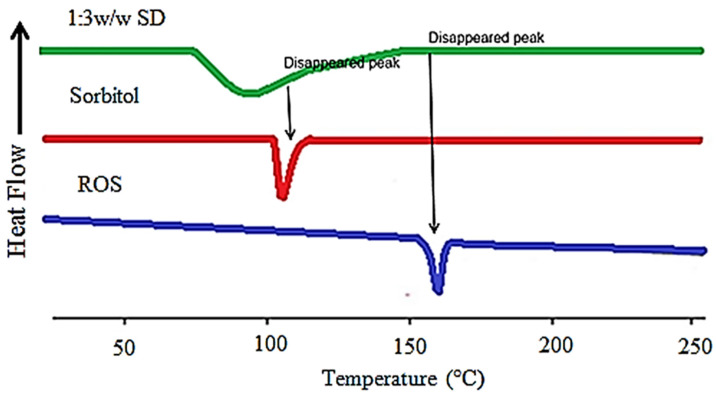
DSC curves of pure ROS, sorbitol, and ROS-SDs (1:3 *w*/*w* SD).

**Figure 4 pharmaceutics-14-01629-f004:**
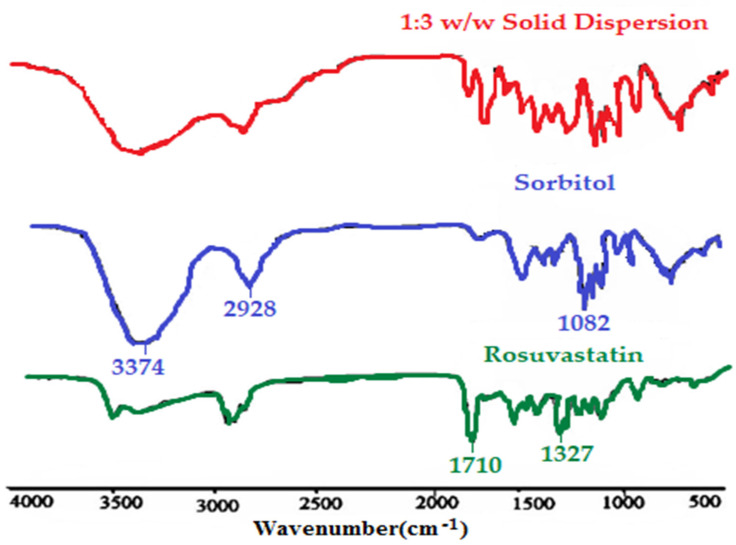
FT-IR spectra of pure ROS, sorbitol, and ROS-SDs (1:3).

**Figure 5 pharmaceutics-14-01629-f005:**
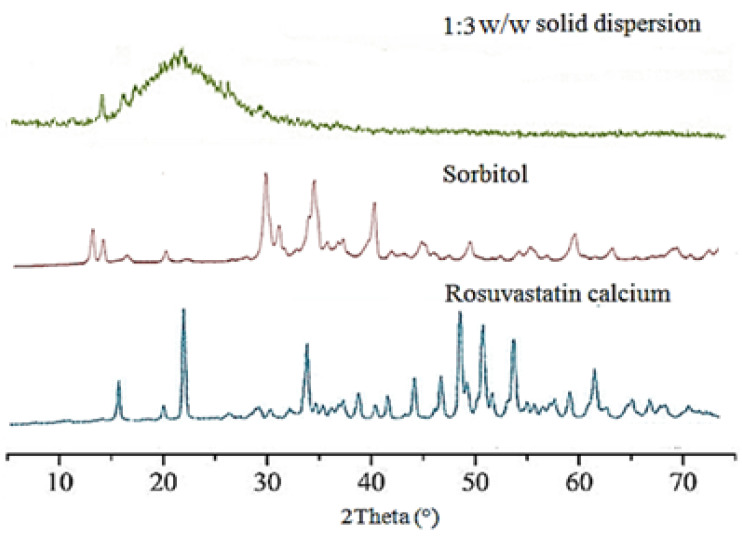
PXRD pattern of pure ROS and ROS-SDs (1:3).

**Figure 6 pharmaceutics-14-01629-f006:**
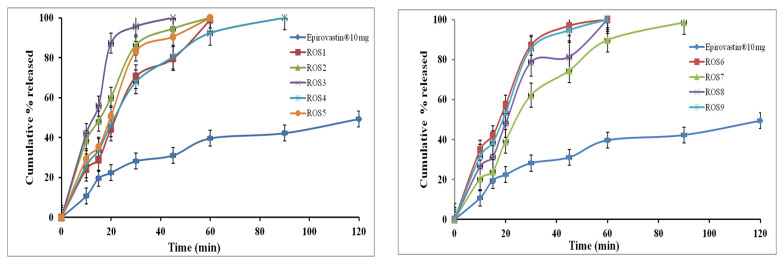
In vitro release profiles of ROS from the commercial and prepared mono tablets (mean ± SD, *n* = 3).

**Figure 7 pharmaceutics-14-01629-f007:**
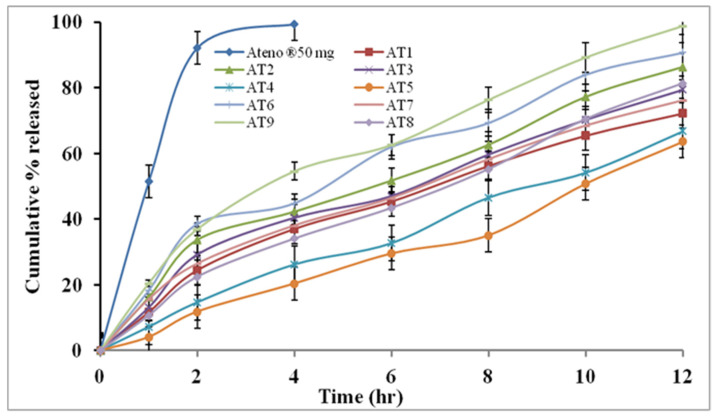
In vitro release profile of AT from the commercial and prepared mono tablets (mean ± SD, *n* = 3).

**Figure 8 pharmaceutics-14-01629-f008:**
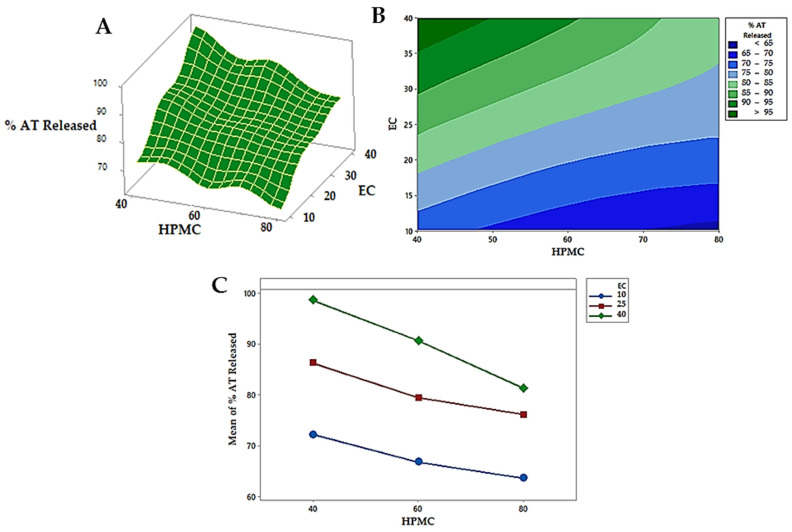
(**A**–**C**) The response surface plot (**A**), contour plots (**B**), and main effects plots (**C**) estimate the effect of independent variables on % released after 12 h.

**Figure 9 pharmaceutics-14-01629-f009:**
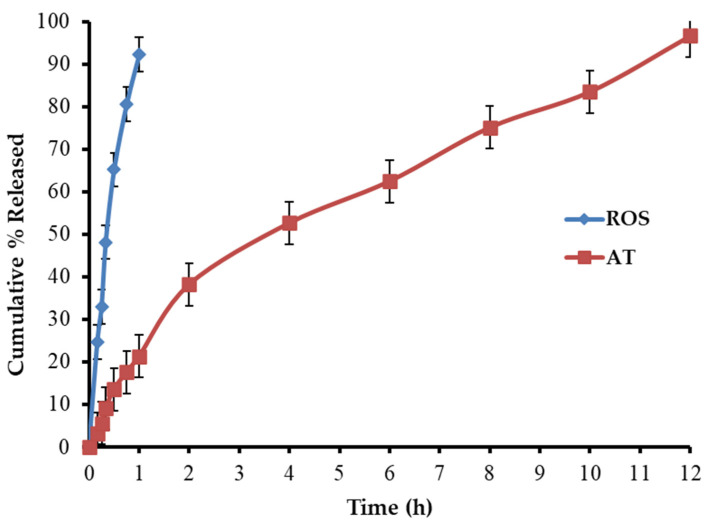
In vitro release profile of ROS and AT from the prepared BLTs (mean ± SD, *n* = 3).

**Table 1 pharmaceutics-14-01629-t001:** Composition of different ROS-IRL mono tablets.

F. Code	CCS	CP	SSG	Lactose Monohydrate
ROS1	9	***	***	97.5
ROS2	12	***	***	94.5
ROS3	15	***	***	91.5
ROS4	***	9	***	97.5
ROS5	***	12	***	94.5
ROS6	***	15	***	91.5
ROS7	***	***	9	97.5
ROS8	***	***	12	94.5
ROS9	***	***	15	91.5

*** = not applicable.

**Table 2 pharmaceutics-14-01629-t002:** Independent factors and response for 3^2^ full factorial design.

Independent Variables	Levels		
	High (+1)	Medium (0)	Low (−1)
HPMC (% *w*/*w*), (*X*_1_)	40	30	20
EC (% *w*/*w*), (*X*_2_)	20	12.5	5
Dependent response		Aim	
% released at 12 h (*Y*_1_)		Maximize	

**Table 3 pharmaceutics-14-01629-t003:** Composition of different AT/SR mono tablets.

AT/SR									
Ingredients (mg)	AT1	AT2	AT3	AT4	AT5	AT6	AT7	AT8	AT9
AT	50	50	50	50	50	50	50	50	50
HPMC K100	40	40	60	60	80	60	80	80	40
EC	10	25	25	10	10	40	25	40	40
Sodium bicarbonate	20	20	20	20	20	20	20	20	20
Mg Stearate	2	2	2	2	2	2	2	2	2
Lactose monohydrate	78	63	43	58	38	28	23	8	48

**Table 4 pharmaceutics-14-01629-t004:** % DE and RDR at different times.

Formula	% DE_30 min_ *, *n* = 3	% DE_60 min_ *, *n* = 3	% DE_120 min_ *, *n* = 3	RDR_30 min_ **, *n* = 3	RDR_60 min_ **, *n* = 3	RDR_120 min_ **, *n* = 3
Pure ROS	5.45 ± 0.708	12.50 ± 1.11	23.98 ± 1.55	1.00 ± 0.152	1.00 ± 0.108	1.00 ± 0.116
SD *** 1:1	16.67 ± 1.75	27.49 ± 1.13	41.07 ± 2.09	2.52 ± 0.426	1.60 ± 0.243	1.55 ± 0.145
SD 1:2	22.57 ± 1.69	39.88 ± 2.15	60.30 ± 2.56	3.43 ± 0.611	2.36 ± 0.780	2.22 ± 0.207
SD 1:3	29.21 ± 2.16	50.31 ± 2.06	70.97 ± 3.23	5.10 ± 0.351	2.82 ± 0.565	2.49 ± 0.233
SD 1:4	29.14 ± 1.84	50.05 ± 1.78	69.99 ± 1.94	5.25 ± 0.444	2.73 ± 0.621	2.42 ± 1.78

* % DE = dissolution efficiency. ** RDR = the relative dissolution rate. *** SD = solid dispersion.

**Table 5 pharmaceutics-14-01629-t005:** Pre-compression characterization parameters.

F. Code	The Angle of Repose (*θ*)	Carr’s Index (%)	Hausner’s Ratio
ROS tablets powder blends			
ROS1	29.23 ± 1.55	16.30 ± 2.23	1.32 ± 0.023
ROS2	24.66 ± 1.29	19.25 ± 1.89	1.21 ± 0.012
ROS3	18.59 ± 1.06	14.84 ± 1.25	1.20 ± 0.007
ROS4	25.83 ± 1.32	16.90 ± 2.08	1.27 ± 0.071
ROS5	22.18 ± 1.66	13.78 ± 1.33	1.30 ± 0.037
ROS6	16.59 ± 1.10	11.72 ± 1.94	1.10 ± 0.055
ROS7	17.82 ± 1.41	12.45 ± 2.61	1.19 ± 0.072
ROS8	27.74 ± 1.20	18.61 ± 1.73	1.28 ± 0.026
ROS9	20.65 ± 1.32	19.04 ± 1.40	1.12 ± 0.047
AT tablets powder blends			
AT1	22.74 ± 1.30	17.42 ± 1.91	1.09 ±0.046
AT2	25.81 ± 1.05	12.84 ± 2.04	1.13 ± 0.109
AT3	19.25 ± 1.22	16.51 ± 2.80	1.22 ± 0.047
AT4	26.49 ± 1.09	10.30 ± 2.87	1.20 ± 0.040
AT5	21.20 ± 1.07	16.33 ± 1.26	1.23 ± 0.030
AT6	28.70 ± 1.32	14.94 ± 1.84	1.11 ± 0.037
AT7	25.23 ± 1.63	11.70 ± 2.84	1.18 ± 0.015
AT8	27.77 ± 1.27	11.94 ± 1.22	1.24 ± 0.052
AT9	20.35 ± 2.14	12.40 ± 1.87	1.19 ± 0.017

**Table 6 pharmaceutics-14-01629-t006:** Post-compression characterization parameters of the prepared mono tablets.

F. Code	Drug Content (%)	Weight (mg)	Thickness (mm)	Friability (%)	Hardness (Kg/cm^2^)
	ROS Immediate-Release Mono Tablets				
ROS1	96.12 ± 4.34	140.85 ± 3.82	1.87 ± 0.34	0.672 ± 0.004	3.85 ± 0.52
ROS2	99.54 ± 2.87	144.43 ± 2.94	1.91 ± 0.56	0.580 ± 0.001	3.25 ± 0.40
ROS3	100.82 ± 3.40	151.24 ± 5.43	2.08 ± 0.81	0.532 ± 0.007	4.60 ± 0.71
ROS4	97.09 ± 1.28	144.56 ± 3.30	1.90 ± 0.24	0.693 ± 0.003	4.09 ± 0.55
ROS5	95.70 ± 4.20	150.00 ± 2.75	1.99 ± 0.62	0.802 ± 0.006	3.95 ± 0.84
ROS6	95.29 ± 2.06	144.14 ± 3.08	1.81 ± 0.11	0.596 ± 0.003	4.58 ± 0.22
ROS7	98.95 ± 3.09	141.74 ± 2.79	1.83 ± 0.26	0.895 ± 0.009	4.45 ± 0.95
ROS8	97.18 ± 1.25	149.10 ± 3.42	1.95 ± 0.61	0.711 ± 0.004	3.54 ± 0.66
ROS9	100.04 ± 2.31	140.72 ± 2.83	1.90 ± 0.56	0.655 ± 0.006	4.11 ± 0.38
	AT Sustained-Release Mono Tablets				
AT1	98.60 ± 2.34	193.56 ± 4.10	2.99 ± 0.37	0.734 ± 0.004	5.60 ± 0.18
AT2	97.50 ± 2.67	195.94 ± 3.81	2.90 ± 0.62	0.631 ± 0.002	5.84 ± 0.23
AT3	99.21 ± 1.90	197.78 ± 2.55	3.09 ± 0.11	0.540 ± 0.005	7.69 ± 0.54
AT4	95.33 ± 2.33	201.40 ± 4.26	2.92 ± 0.20	0.628 ± 0002	6.52 ± 0.12
AT5	98.10 ± 2.84	189.67 ± 2.38	3.01 ± 0.73	0.583 ± 0.005	7.72 ± 0.40
AT6	100.42 ± 1.25	202.62 ± 5.73	3.11 ± 0.74	0.555 ± 0.008	6.66 ± 0.31
AT7	100.11 ± 2.69	199.50 ± 2.85	2.95 ± 0.26	0.773 ± 0.003	7.15 ± 0.27
AT8	96.85 ± 2.55	204.93 ± 5.20	3.00 ± 0.18	0.730 ± 0.004	6.50 ± 0.16
AT9	96.35 ± 1.62	200.21 ± 5.83	3.05 ± 0.34	0.749 ± 0.001	5.98 ± 0.21

**Table 7 pharmaceutics-14-01629-t007:** Buoyancy study of AT-SR mono tablets.

F. Code	Buoyancy Lag Time (min)	Total Floating Time (h)
AT1	4.42 ± 0.21	22.41 ± 1.20
AT2	5.53 ± 0.32	23.12 ± 0.80
AT3	2.82 ± 0.08	18.91 ± 1.14
AT4	2.10 ± 0.11	17.54 ± 1.31
AT5	1.20 ± 0.15	13.83 ± 0.74
AT6	3.75 ± 0.41	20.10 ± 0.89
AT7	1.53 ± 0.16	14.95 ± 1.08
AT8	2.08 ± 0.24	15.18 ± 1.14
AT9	6.35 ± 0.83	24.01 ± 1.91

**Table 8 pharmaceutics-14-01629-t008:** Analysis of variance for the % released after 12 h.

Source	Sum of Squares	Df	Mean Square	F-Ratio	*p*-Value
A:HPMC	230.64	1	230.64	167.94	0.0010
B:EC	772.935	1	772.935	562.82	0.0002
AA	1.62	1	1.62	1.18	0.3569
AB	19.36	1	19.36	14.10	0.0330
BB	8.405	1	8.405	6.12	0.0898
Total error	4.12	3	1.37333		
Total correction	1037.08	8			

HPMC = hydroxypropyl methylcellulose. EC = ethylcellulose. AA = quadratic effect of HPMC. BB = quadratic effect of EC. AB = interactive effect of HPMC and EC.

**Table 9 pharmaceutics-14-01629-t009:** Kinetic analysis of the prepared IRT/SRT formulations.

F. Code	Zero-Order		First-Order		Higuchi-Diffusion		n
	r^2^	K_0_	r^2^	K_1_	r^2^	K_H_	
Kinetics of ROS-Ca^2+^-IRT							
ROS1	0.958	0.593	0.981	0.010	0.979	7.136	…..
ROS2	0.307	0.284	0.636	0.048	0.114	1.247	…..
ROS3	0.459	0.443	0.526	0.038	0.307	3.496	…..
ROS4	0.970	0.534	0.990	0.008	0.982	6.376	…..
ROS5	0.956	0.851	0.983	0.038	0.976	0.239	…..
ROS6	0.099	0.087	0.811	0.049	0.112	1.159	…..
ROS7	0.969	0.459	0.983	0.006	0.978	5.460	…..
ROS8	0.969	0.755	0.991	0.019	0.980	9.001	…..
ROS9	0.312	0.280	0.896	0.051	0.480	5.081	…..
Kinetics of AT-SRT							
AT1	0.964	5.769	0.982	0.103	0.995	21.65	0.703
AT2	0.958	6.593	0.960	0.150	0.980	24.76	0.621
AT3	0.960	6.167	0.981	0.122	0.992	23.14	0.6719
AT4	0.994	5.36	0.977	0.086	0.941	19.39	…..
AT5	0.988	5.08	0.948	0.077	0.898	18.01	…..
AT6	0.941	7.01	0.966	0.183	0.985	26.6	0.602
AT7	0.966	5.71	0.987	0.112	0.991	22.22	0.619
AT8	0.989	6.47	0.958	0.129	0.956	23.64	…..
AT9	0.949	7.59	0.842	0.297	0.992	28.81	0.606

**Table 10 pharmaceutics-14-01629-t010:** AT/ROS BLTs stability study profile.

Parameters/Time		Initial Time	1 Month	2 Months	4 Months	6 Months
Drug content (%)	AT	99.65 ± 1.6	99.20 ± 2.3	98.48 ± 1.0	97.77 ± 1.7	96.80 ± 2.4
	ROS	98.54 ± 2.3	98.11 ± 1.8	97.61 ± 1.4	96.33 ± 3.2	95.16 ± 1.7
Average weight (mg)		347.51 ± 4.1	347.44 ± 3.7	347.02 ± 4.2	346.79 ± 4.3	346.54 ± 2.6
Hardness (kg/cm^2^)		6.43 ± 0.2	6.47 ± 0.3	6.25 ± 0.2	6.17 ± 0.8	6.20 ± 0.2
Cumulative % drug released	ROS at 60 min	92.34 ± 2.2	92.49 ± 1.4	92.33 ± 2.0	91.55 ± 1.5	91.70 ± 1.2
	AT at 12 h	96.65 ± 3.3	96.16 ± 1.3	96.27 ± 2.0	96.10 ± 1.0	96.39 ± 2.1

**Table 11 pharmaceutics-14-01629-t011:** Changes in lipid profile (mean values ± SE, *n* = 3) in rabbits in all groups over all the experimental periods.

Time (Week)	Groups	Cholesterol (mg/dL)	TG (mg/dL)	HDL-c (mg/dL)	LDL-c (mg/dL)
Week 0	A (control)	74.38 ± 0.61 ^d^	51.37 ± 0.92 ^c^	22.37 ± 0.80 ^a^	41.72 ± 0.29 ^c^
	B (HFD)	71.14 ± 0.84 ^d^	48.14 ± 0.20 ^c^	19.14 ± 0.97 ^a^	42.37 ± 0.47 ^c^
	C (HFD)	75.15 ± 0.39 ^d^	52.15 ± 0.19 ^c^	23.15 ± 0.82 ^a^	41.57 ± 0.51 ^c^
Week 4	A (control)	76.11 ± 0.15 ^d^	53.11 ± 0.48 ^c^	24.11 ± 0.41 ^a^	41.37 ± 0.62 ^c^
	B (HFD)	82.54 ± 0.48 ^cd^	59.54 ± 0.39 ^c^	19.27 ± 0.73 ^a^	51.36 ± 0.54 ^b^
	C (HFD)	83.78 ± 0.67 ^cd^	60.78 ± 0.71 ^c^	18.74 ± 0.27 ^b^	52.88 ± 0.67 ^b^
Week 8	A (control)	70.72 ± 0.87 ^d^	47.72 ± 0.52 ^c^	25.49 ± 0.19 ^a^	35.68 ± 0.27 ^c^
	B (HFD)	93.87 ± 0.59 ^c^	70.87 ± 0.97 ^b^	20.58 ± 0.30 ^b^	59.11 ± 0.19 ^b^
	C (HFD)	95.73 ± 0.74 ^c^	72.73 ± 0.74 ^b^	19.24 ± 0.43 ^b^	61.94 ± 0.24 ^b^
Week 10	A (control)	76.81 ± 1.25 ^d^	53.81 ± 0.29 ^c^	26.28 ± 0.81 ^a^	39.76 ± 0.35 ^c^
	B (HFD)	101.27 ± 0.91 ^b^	78.27 ± 0.63 ^b^	18.25 ± 0.37 ^b^	67.36 ± 0.64 ^b^
	C (HFD)	107.67 ± 0.28 ^b^	84.67 ± 0.18 ^b^	17.23 ± 0.23 ^b^	73.50 ± 0.74 ^b^
Week 12	A (control)	74.34 ± 1.23 ^d^	51.34 ± 0.34 ^c^	26.37 ± 0.12 ^a^	37.70 ± 0.67 ^c^
	B (HFD)	125.74 ± 0.28 ^a^	102.74 ± 0.41 ^a^	15.27 ± 0.28 ^c^	89.92 ± 0.42 ^b^
	C (HFD)	128.22 ± 0.36 ^a^	105.22 ± 0.59 ^a^	14.79 ± 0.71 ^c^	92.38 ± 0.18 ^b^
Week 14	A (control)	78.59 ± 0.71 ^d^	55.59 ± 0.37 ^c^	26.59 ± 0.60 ^a^	40.88 ± 0.17 ^c^
	B (HFD)	138.45 ± 0.29 ^a^	115.44 ± 0.82 ^a^	14.23 ± 0.37 ^c^	101.13 ± 0.28 ^a^
	C (HFD) + bilayer	119.91 ± 0.20 ^b^	96.91 ± 0.24 ^ab^	17.29 ± 0.81 ^b^	83.23 ± 0.39 ^b^
Week 15	A (control)	77.18 ± 0.89 ^d^	54.18 ± 0.37 ^c^	25.33 ± 0.39 ^a^	41.01 ± 0.71 ^c^
	B (HFD)	151.67 ± 0.87 ^a^	128.67 ± 0.42 ^a^	13.28 ± 0.19 ^c^	112.65 ± 0.80 ^a^
	C (HFD) + bilayer	105.29 ± 0.19 ^b^	82.29 ± 0.57 ^b^	18.29 ± 0.47 ^b^	70.54 ± 0.76 ^b^
Week 16	A (control)	71.25 ± 0.38 ^d^	58.81 ± 0.38 ^c^	27.29 ± 0.38 ^a^	38.19 ± 0.89 ^c^
	B (HFD)	174.48 ± 0.34 ^a^	151.48 ± 0.41 ^a^	11.17 ± 0.67 ^c^	133.01 ± 0.27 ^a^
	C (HFD) + bilayer	94.89 ± 0.49 ^cd^	71.89 ± 0.08 ^bc^	21.23 ± 0.89 ^a^	59.28 ± 0.63 ^bc^

TG = triglycerides, HDL-c = high density lipoprotein cholesterol, LDL-c = low density lipoprotein cholesterol, HFD = high-fat diet, bilayer: (ROS 10 mg and AT 50 mg)/kg b.wt. Means in the same column followed by different letters were significantly different at *p* < 0.05.

**Table 12 pharmaceutics-14-01629-t012:** Changes in the blood pressure, heart rate, and body weight (mean values ± SE, *n* = 3) in rabbits in all groups over all the experimental periods.

Time (Week)	Groups	Blood Pressure (mmHg)	Heart Rate (Number/Minute)	Body Weight (gm)
Week 0	A (control)	75.83 ± 1.31 ^d^	112.14 ± 0.87 ^c^	1081.80 ± 0.91 ^d^
	B (HFD)	78.41 ± 2.24 ^d^	118.36 ± 0.97 ^c^	1117.13 ± 1.51 ^d^
	C (HFD)	77.15 ± 1.69 ^d^	112.87 ± 0.71 ^c^	1095.48 ± 1.09 ^d^
Week 4	A (control)	80.31 ± 2.27 ^d^	115.21 ± 1.91 ^c^	1321.47 ± 1.38 ^d^
	B (HFD)	95.79 ± 1.94 ^cd^	135.54 ± 1.47 ^c^	1647.21 ± 0.56 ^c^
	C (HFD)	95.96 ± 2.37 ^cd^	137.78 ± 1.38 ^c^	1660.49 ± 2.17 ^c^
Week 8	A (control)	74.87 ± 1.12 ^d^	121.58 ± 1.74 ^c^	1551.29 ± 2.35 ^c^
	B (HFD)	93.87 ± 0.59 ^c^	142.31 ± 2.63 ^b^	2259.28 ± 3.18 ^b^
	C (HFD)	95.73 ± 0.74 ^c^	151.87 ± 3.17 ^b^	2191.37 ± 2.97 ^b^
Week 10	A (control)	81.24 ± 2.19 ^d^	118.38 ± 1.78 ^c^	1628.36 ± 3.46 ^c^
	B (HFD)	111.57 ± 2.35 ^b^	156.38 ± 2.17 ^b^	2561.92 ± 3.48 ^a^
	C (HFD)	113.17 ± 1.63 ^b^	161.31 ± 3.29 ^b^	2449.39 ± 2.37 ^a^
Week 12	A (control)	82.91 ± 2.47 ^d^	122.11 ± 2.64 ^c^	1711.14 ± 2.19 ^c^
	B (HFD)	123.29 ± 3.12 ^a^	172.25 ± 3.57 ^a^	2778.32 ± 3.08 ^a^
	C (HFD)	132.46 ± 2.48 ^a^	166.39 ± 2.78 ^a^	1760.27 ± 3.91 ^a^
Week 14	A (control)	81.58 ± 2.84 ^d^	115.71 ± 3.26 ^c^	1817.07 ± 2.49 ^c^
	B (HFD)	141.12 ± 3.38 ^a^	185.68 ± 4.15 ^a^	2892.33 ± 4.17 ^a^
	C (HFD) + bilayer	122.17 ± 2.67 ^b^	159.81 ± 3.38 ^ab^	2423.01 ± 2.93 ^b^
Week 15	A (control)	84.55 ± 3.19 ^d^	117.27 ± 3.84 ^c^	1895.33 ± 3.61 ^c^
	B (HFD)	148.34 ± 2.49 ^a^	188.38 ± 4.31 ^a^	3012.14 ± 4.74 ^a^
	C (HFD) + bilayer	109.37 ± 3.28 ^b^	145.37 ± 3.42 ^b^	2101.15 ± 1.78 ^b^
Week 16	A (control)	81.67 ± 3.19 ^d^	114.32 ± 2.78 ^c^	1937.92 ± 2.48 ^c^
	B (HFD)	165.34 ± 4.27 ^a^	192.67 ± 4.59 ^a^	3130.39 ± 4.37 ^a^
	C (HFD) + bilayer	91.27 ± 2.38 ^cd^	136.47 ± 3.24 ^bc^	2071.36 ± 2.84 ^bc^

HFD = high-fat diet, bilayer = (ROS 10 mg and AT 50 mg)/kg b.wt. Means in the same column followed by different letters were significantly different at *p* < 0.05.

## Data Availability

The datasets generated during and/or analyzed during the current study are available from the corresponding authors on reasonable request.

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
