# Peer review of "Tailoring of Rosuvastatin Calcium and Atenolol Bilayer Tablets for the Management of Hyperlipidemia Associated with Hypertension: A Preclinical Study"

_pharmaceutics, 2022, doi:10.3390/pharmaceutics14081629_

Round 1

Reviewer 1 Report

The article entitled "Tailoring of Rosuvastatin Calcium and Atenolol Bilayer Tablets for the Management of Hyperlipidemia Associated with Hypertension: A Preclinical Study" presents a very complete study, and should be accepted in this journal, however some suggestions are listed below to suit the article.

Carefully check the English to avoid expressions like..." have has"...

In item 2.2 was methanol used, even considering its waste evaporation to be problematic for a larger scale production? Wouldn't it be possible to use another solvent?

Replace all concentration units with mol/L or g/L.

In item 2.3 (dissolution) you must indicate the pH of the system.

Replace Thermograms by DSC curves”, and why did you only use temperature up to 200oC?

The FTIR description needs to be better described.

In Table 1, only the items that were varied should remain, the constants do not need to be.

The presence of sodium bicarbonate cannot interfere with the pressure of future patients, as we know that sodium is a complicating factor. Because they didn't use the bicarbonate of another cation.

 For the equations indicated in sub-item 2.9.7, it is necessary to cite the original reference for each model.

For in vivo tests, it is necessary to cite data from the ethics committee protocol.

The results should start with the characterization of the systems, not with the release.

The DSC data would need to be redone with a heating rate of 2 or 5oC, as for the 1:3 system there appear to be two events in the region between 80-150oC. The change from crystalline to amorphous would need the help of an X-ray diffraction for a more secure statement.

For the FTIR, the sentence is presented: "All these peaks were well observed with an insignificant shifting in the spectrum of the ROS-SDs formula to confirm that the absence of interaction between ROS and sorbitol." However, a detailed analysis of these bands is necessary. .

Further discussion is needed, as several items 3.1.4, 3.2 and others are presented with the results but lack of a discussion of the analysis of the results.

Why in item 3.4 the faster release ROS3 system was sought, wouldn't a more controlled release be more interesting?

Characterizations were missing for systems other than those presented in item 3.1

The “In vitro Dissolution Investigation of BLTs” tests were performed at pH 1.2 for a very long time, but the correct thing was to do it at this pH for a time of 2 hours and then adjust the pH to 7.4. Thus, it is necessary to redo this test with these conditions.

Author Response

Reviewer 1

The article entitled "Tailoring of Rosuvastatin Calcium and Atenolol Bilayer Tablets for the Management of Hyperlipidemia Associated with Hypertension: A Preclinical Study" presents a very complete study, and should be accepted in this journal, however, some suggestions are listed below to suit the article.

  1. Comment 1 from Reviewer 1: Carefully check the English to avoid expressions like..." have has"...
  • Response to reviewer: Thank you for the suggestion. The English has been checked and the manuscript revised carefully.
  1. Comment 2 from Reviewer 1: In item 2.2 was methanol used, even considering its waste evaporation to be problematic for a larger scale production? Wouldn't it be possible to use another solvent?
  • Response to the reviewer:

We used methanol, because of its low vapor pressure, so rapid evaporation occurs.

For a larger scale production, the distillation of the solvent to recover methanol can be used for recycling and limiting environmental hazards.

  1. Comment 3 from Reviewer 1: Replace all concentration units with mol/L or g/L.
  • Response to the reviewer:

We used mass percentage (% w/w) to express the proportion of a particular substance within a powder mixture, measured by weight.

  1. Comment 4 from Reviewer 1: In item 2.3 (dissolution) you must indicate the pH of the system.
  • Response to the reviewer:

The pH of the system has been added.

  1. Comment 5 from Reviewer 1: Replace Thermograms by DSC curves”, and why did you only use temperature up to 200 oC?
  • Response to reviewer:

-The Thermograms have been replaced by DSC curves.

-The thermal behavior was studied in temperature ranges of 25 – 250 °C, but we made a typo.

  1. Comment 6 from Reviewer 1: The FTIR description needs to be better described.
  • Response to the reviewer:

The English have been modified.

  1. Comment 7 from Reviewer 1: In Table 1, only the items that were varied should remain, the constants do not need to be.
  • Response to the reviewer:

The constants have been removed.

  1. Comment 8 from Reviewer 1:The presence of sodium bicarbonate cannot interfere with the pressure of future patients, as we know that sodium is a complicating factor. Because they didn't use the bicarbonate of another cation.
  • Response to the reviewer:

The amount of sodium bicarbonate per tablet is very small (20mg), such an amount containing 0.005 mg of sodium had no effect on blood pressure. Excessive intake of sodium > 5 g per day as defined by the World Health Organization has been shown to significantly increase blood pressure.

Reference: Grillo, A., Salvi, L., Coruzzi, P., Salvi, P., & Parati, G. (2019). Sodium intake and hypertension. Nutrients11(9), 1970.‏

  1. Comment 9 from Reviewer 1:For the equations indicated in sub-item 2.9.7, it is necessary to cite the original reference for each model.
  • Response to the reviewer:

The original reference for each model has been added

  1. Comment 10 from Reviewer 1:For in vivo tests, it is necessary to cite data from the ethics committee protocol.
  • Response to the reviewer: Ethical committee approval added.
  1. Comment 11 from Reviewer 1:The results should start with the characterization of the systems, not with the release.
  • Response to the reviewer:

We started with release to determine which ratio of solid dispersion has high dissolution parameters to be selected for further analysis.

  1. Comment 12 from Reviewer 1: The DSC data would need to be redone with a heating rate of 2 or 5oC, as for the 1:3 system there appear to be two events in the region between 80-150oC. The change from crystalline to amorphous would need the help of an X-ray diffraction for a more secure statement.
  • Response to the reviewer:

In response to the valuable comment, we conduct XRD study and add data with a discussion to the manuscript,

2.5. Analysis of powder X-ray diffraction (PXRD)

An automated X-ray diffractometer Philips 1710 was used to analyze the different samples (pure rosuvastatin calcium, sucrose, and ROS-SDs, 1:3 w/w). CuKα radiation at 40 kV and 30 mA (l Kα = 1.4309 Å) was used to detect diffraction peaks. At a scanning speed of 5°/minute, the samples under investigation were scanned from 3° to 70° [30].

3.1.4. Powder X-ray diffraction analysis

The PXRD diffractograms that were produced showed that pure ROS exhibited character-istic peaks in the frequencies 2θ =16.79, 23.14, and 34.06. These peaks indicate that pure ROS is crystalline in composition. However, in treated ROR powder (ROS-SDs, 1:3 w/w), both the height of the peaks and the number of peaks were reduced, which indicated that the treated ROR powder had a relatively low crystallinity.

These ROS-SDs (1:3 w/w) have decreased ROS crystallinity, as evidenced by the lower peak heights and the disappearance of some significant peaks in their PXRD patterns (Figure 5). In the ROS-SDs (1:3 w/w) studied, ROS was found to have transformed from crystalline to amorphous form [61,62].

We agree with the reviewer that, with a heating rate of 2 or 5oC, as for the 1:3 system there appear to be two events in the region between 80-150oC, but, unfortunately, we have no more samples to conduct a new DSC  experiment.

  1. Comment 13 from Reviewer 1: For the FTIR, the sentence is presented: "All these peaks were well observed with an insignificant shifting in the spectrum of the ROS-SDs formula to confirm that the absence of interaction between ROS and sorbitol." However, a detailed analysis of these bands is necessary.
  • Response to the reviewer:

The bands have been added

  1. Comment 14 from Reviewer 1: Further discussion is needed, as several items 3.1.4, 3.2, and others are presented with the results but lack of a discussion of the analysis of the results.
  • Response to the reviewer:

The indication of results has been added.

  1. Comment 15 from Reviewer 1: Why in item 3.4 the faster release ROS3 system sought, wouldn't a more controlled release be more interesting?
  • Response to the reviewer:

The goal of the study was to increase the bioavailability of rosuvastatin by improving its dissolution rather than controlling its release, as it has an elimination half-life of 20 hours and is already administered once daily.

  1. Comment 16 from Reviewer 1: Characterizations were missing for systems other than those presented in item 3.1
  • Response to the reviewer:

Characterization of rosuvastatin solid dispersion (3.1) includes dissolution, DSC, IR. we rearranged the numbers.

  1. Comment 17 from Reviewer 1: The “In vitro Dissolution Investigation of BLTs” tests were performed at pH 1.2 for a very long time, but the correct thing was to do it at this pH for a time of 2 hours and then adjust the pH to 7.4. Thus, it is necessary to redo this test with these conditions.
  • Response to the reviewer:

For rosuvastatin, ROS3 showed 100% drug release in the simulated gastric fluid within 45 minutes. In Atenolol, the total floating time is greater than 12hr as observed in table 7 to extend the residence time of the drug in the stomach and delay the drug release. So no need to change the dissolution medium.

Reviewer 2 Report

This manuscript deals with bilayer tablets composed of a gastro retentive layer for sustained release of atenolol and an immediate release layer for improved solubility of rosuvastatin calcium. In-vitro and in-vivo analysis have been performed on these devices.

General remarks: Please check numeration of sections (for example section 2.3 exists twice) and correct the numerous typos. The paper need to be revised to add clarifications, an example: The equipment used is not always (fully) indexed as follows, type of equipment, supplier, city, nation.

The discussions of the obtained results are not exhaustive.

Why were tablet properties (friability, hardness, uniformity of weight, etc) examined on single layer tablets, only? Instead it would have been interesting to see if the bilayer tablets correspond to the requirements of the pharmacopeia and emit the necessary dissolution behavior.

Here some detailed remarks classified by the sections:

Introduction :

Please include some references justifying the employment of the rabbit model in the preclinical study.

Materials & Methods:

2.1: What kind of ethylcellulose was used (degree of substitution, Molweight and so on)?

In section 2.2 you describe the preparation of the solid dispersions. What was the residual solvent content within the obtained powders?

Section 2.3 “Dissolution Study of the Prepared ROS-SDs (1:3 w/w)” needs more clarification. Was an equivalent to 10mg pure ROS used with the different ROS-SDs? Were the samples taken filtered, if so what filter was used? In dissolution studies of poorly soluble drugs and their solid dispersions drug absorption is often observed. How was drug precipitation avoided within the samples (co-solvents, dilution, surfactants)?

2.5: What type of punches was used (shape, diameter)?

The tablet weight variation study was conducted according to the USP specifications (section 2.9.2)? If so, please add it to the text.

2.7:  Table 3 Could you please add AT1…etc to the naming, as in the discussion section you refer to the formulations as A1…

In section 2.10 you describe the preparation of the bilayer tablets. What was the compression force used in the 1st and second layer compression, respectively? What was the optimized formula of AT?

2.13: What does SEM stand for?

Results and discussion:

3.1.1:  How is the standard deviation between ROS-SD 1:3 and ROS-SD 1:4 influencing the DE or RDR? Is there a significant difference in dissolution efficacy between the two solid dispersion ratios?

Isn’t it recommended by Khan (1975) to apply the DE expression on time intervals (DE30, 60 and 120, in this case)?

3.1.2: What was the standard deviation between the three samples analyzed (n=3)?

3.1.4: What is the explanation for the increase in flowability from ROS 1 to ROS 2 & 3 and ROS 5 compared to 6 & 7? I would have imagined that the higher the lactose content in the powder blen, the better the flowability, but the contrary seems to be the case here.

3.2: Could you add some discussion, please? Are the values of post-compression within the specifications (requirements USP, e.g.)?

3.4.2 Please format all equations (in italic or eq.) to increase the comprehension of the study. Howdo you explain the fact that increasing the amount of the hydrophobic compound ethylcellulose leads to an increase in rug release? Are there any publications existing that could support your findings? If you would have given some information on the ethylcellulose used, it might be easier to comprehend.

3.6.1 What was the composition of the AT layer? Do you observe any modification in drug release compared to the single layer tablets? If so, please discuss!

3.7. The rosuvastatin release seems to slightly decrease during storage, do you have any information on the tablet’s stability in storage in non-sealed containers? Is there some information on this type of solid dispersion in the literature?

3.8. Was there a negative control group (like group C but not treated)? Group B and C did not have the same diets, as I understand.

Author Response

Reviewer 2

This manuscript deals with bilayer tablets composed of a gastro retentive layer for sustained release of atenolol and an immediate release layer for improved solubility of rosuvastatin calcium. In-vitro and in vivo analysis have been performed on these devices.

General remarks:

  1. Comment 1 from Reviewer 2: Please check the numeration of sections (for example section 2.3 exists twice) and correct the numerous typos. The paper needs to be revised to add clarifications, an example: The equipment used is not always (fully) indexed as follows, type of equipment, supplier, city, and nation.
  • Response to reviewer:

The number of sections has been corrected.

Equipment data has been completed.

  1. Comment 2 from Reviewer 2: The discussions of the obtained results are not exhaustive.
  • Response to reviewer:

Upon the reviewer's comment, we modify and add the discussion

  1. Comment 3 from Reviewer 2: Why were tablet properties (friability, hardness, uniformity of weight, etc) examined on single-layer tablets, only? Instead, it would have been interesting to see if the bilayer tablets correspond to the requirements of the pharmacopeia and emit the necessary dissolution behavior.
  • Response to reviewer:

-We performed a post-compression study on each monolayer tablet to confirm that each was suitable for further study. Then we selected the best formulations of monolayer tablets based on the suitable dissolution pattern.

-We already repeated the tests on the bilayer tablets in section 3.7.

Here are some detailed remarks classified by the sections:

Introduction :

  1. Comment 4 from Reviewer 2: Please include some references justifying the employment of the rabbit model in the preclinical study.
  • Response to the reviewer:

Thank you for the suggestion, we add ref, 26, and 27.

Materials & Methods:

  1. Comment 5 from Reviewer 2: 2.1: What kind of ethylcellulose was used (degree of substitution, Molweight and so on)?
  • Response to reviewer:

The data has been presented.

  1. Comment 6 from Reviewer 2: In section 2.2 you describe the preparation of the solid dispersions. What was the residual solvent content within the obtained powders?
  • Response to the reviewer:

We dried until constant weight obtained to ensure that there is no solvent residue.

  1. Comment 7 from Reviewer 2: Section 2.3 “Dissolution Study of the Prepared ROS-SDs (1:3 w/w)” needs more clarification. Was an equivalent to 10mg pure ROS used with the different ROS-SDs? Were the samples taken filtered, if so what filter was used? In dissolution studies of poorly soluble drugs and their solid dispersions drug absorption is often observed. How was drug precipitation avoided within the samples (co-solvents, dilution, surfactants)?
  • Response to the reviewer:

-Solid dispersions equivalent to 10mg pure ROS were used for dissolution studies.

-The samples were filtered through 0.22 μm membrane filter.

- We didn't observe any precipitation in samples before measurement.

  1. Comment 8 from Reviewer 2: 2.5: What type of punches was used (shape, diameter)?
  • Response to the reviewer:

Flat faced 10mm punches

  1. Comment 9 from Reviewer 2: The tablet weight variation study was conducted according to the USP specifications (section 2.9.2)? If so, please add it to the text.
  • Response to the reviewer:

The sentence has been added.

  1. Comment 10 from Reviewer 2: 2.7:  Table 3 Could you please add AT1…etc to the naming, as in the discussion section you refer to the formulations as A1…
  • Response to the reviewer:

AT1 to AT9 have been added.

  1. Comment 11 from Reviewer 2: In section 2.10 you describe the preparation of the bilayer tablets. What was the compression force used in the 1stand second layer compression, respectively? What was the optimized formula of AT?
  • Response to the reviewer:

-The first layer (sustained-release layer) was compressed at 2-4 kg/cm2, then the second layer (fast-release portion) was added directly onto the obtained tablet, and then recompressed together at 6-8 kg/cm2 to combine them.

- The optimized formula of AT is that contains HPMC (20.2 %) and EC (19.96 %) to release 98.1 % of AT within 12 hr as described in section 3.5.2.

  1. Comment 12 from Reviewer 2: 2.13: What does SEM stand for?
  • Response to the reviewer:

 Many thanks for the reviewer comment, we edit this typo error

Results and discussion:

  1. Comment 13 from Reviewer 2: 3.1.1:  How is the standard deviation between ROS-SD 1:3 and ROS-SD 1:4 influencing the DE or RDR? Is there a significant difference in dissolution efficacy between the two solid dispersion ratios? Isn’t it recommended by Khan (1975) to apply the DE expression on time intervals (DE3060and 120, in this case)?
  • Response to reviewer:

- In the calculation of DE and RDR, we took the average value of the data (mean).

-The results showed that there is no significant difference in dissolution parameters between the two solid dispersion ratios.

- DE and RDR were calculated at different time intervals in table 4. 

  1. Comment 14 from Reviewer 2: 3.1.2: What was the standard deviation between the three samples analyzed (n=3)?
  • Response to the reviewer:

Many thanks for the reviewer comment, we edit this typo error

  1. Comment 15 from Reviewer 2: 3.1.4: What is the explanation for the increase in flowability from ROS 1 to ROS 2 & 3 and ROS 5 compared to 6 & 7? I would have imagined that the higher the lactose content in the powder blend, the better the flowability, but the contrary seems to be the case here.
  • Response to the reviewer:

The numbers are variable in Angle of repose, Carr's index (%), and Hausner's ratio. The order of results is expected to be as follows ROS7, ROS1 then ROS2, ROS5 then ROS3, ROS6 as you said, but we think that the difference in lactose amount is very small to cause a significant effect on flowability.

  1. Comment 16 from Reviewer 2: 3.2: Could you add some discussion, please? Are the values of post-compression within the specifications (requirements USP, e.g.)?
  • Response to the reviewer:

Information has been added.

  1. Comment 17 from Reviewer 2: 3.4.2 Please format all equations (in italic or eq.) to increase the comprehension of the study. How do you explain the fact that increasing the amount of the hydrophobic compound ethylcellulose leads to an increase in rug release? Are there any publications existing that could support your findings? If you would have given some information on the ethylcellulose used, it might be easier to comprehend.
  • Response to the reviewer:

-The equations are formatted in italic.

- The effect of EC was explained and the publication was added.

  1. Comment 18 from Reviewer 2: 3.6.1 What was the composition of the AT layer? Do you observe any modification in drug release compared to the single-layer tablets? If so, please discuss!
  • Response to the reviewer:

-The AT layer contains the optimum concentrations of HPMC (40.4 mg) and EC (39.9 mg) which are obtained by the design software to provide the suitable release pattern (sustained-release).

- No significant modification in drug release compared to the single layer tablets of AT.

  1. Comment 19 from Reviewer 2: 3.7. The rosuvastatin release seems to slightly decrease during storage, do you have any information on the tablet’s stability in storage in non-sealed containers? Is there some information on this type of solid dispersion in the literature?
  • Response to reviewer:

-We believe that there was no significant change in the release of rosuvastatin, this slight decrease in drug release may be due to a handling error or the drug content varies slightly from tablet to tablet.

- no information about the effect of sorbitol on drug stability in the literature.

  1. Comment 20 from Reviewer 2: 3.8. Was there a negative control group (like group C but not treated)? Group B and C did not have the same diets, as I understand.
  • Response to the reviewer:

Group A, was fed a standard diet; Group B, was given a standard diet containing 0.5% cholesterol and 3% soybean oil for 16 weeks without treatment (a negative control group); Group C (a positive control group), was received the same diet of group B for 12 weeks followed by oral treatment with BLTs (10 mg of ROS and 50 mg of AT)/kg b.wt with normal diet for 4 weeks. In section 2.13.2

Reviewer 3 Report

Dear Authors,

Re: Manuscript ID: pharmaceutics-1803396

Title: "Tailoring of Rosuvastatin Calcium and Atenolol Bilayer Tablets for the Management of Hyperlipidemia Associated with Hypertension: A Preclinical Study"

The article describes preparation of rosuvastatin calcium and atenolol bilayer tablets to treat coexisting dyslipidemia and hypertension employing a single product. 

Please find my comments / suggestions below:

1. In the Abstract the sentence containing the expression: " ... to maximize the percent released of AT for 12 h ..." needs correction.

2. Same with the sentence containing: " ... tablets was, respectively found to obey the ...".

3. Animal handling licence number is missing.

4. In Introduction: Please check the sentence ending with: " ... gastrointestinal complications[10-12]." (a gap required  before citations).

5. Same with the sentence: "... The objective of the present work to formulate ...".

6. The main difference(s) of the present work with prior literature need to be clearly mentioned in the Introduction and Conclusion.

7. Please correct the following sentence (in Introduction):  "The formulated tablets were evaluated as illustrated in a Figure 2). ".

8. Also, in the above sentence a bracket is missing and number 2 should be changed to 1 (it refers to Figure 1 not Figure 2).

9. The quality of Figures 3 and 7 need to be improved.

10. In my opinion Figure legends need to be relatively self explanatory and abbreviations such as AT  and  ROS-SDs need to be explained.

11. Several typing mistakes are present in Conclusion. For instance:

... management is typed "mangmant"! ... double "of" is used in the last sentence ... "... rabbits ... which was treated ...", ...

Author Response

Reviewer 3

Comments and Suggestions for Authors

Dear Authors,

Re: Manuscript ID: pharmaceutics-1803396

Title: "Tailoring of Rosuvastatin Calcium and Atenolol Bilayer Tablets for the Management of Hyperlipidemia Associated with Hypertension: A Preclinical Study"

 The article describes preparation of rosuvastatin calcium and atenolol bilayer tablets to treat coexisting dyslipidemia and hypertension employing a single product. 

Please find my comments/suggestions below:

  1. Comment 1 from Reviewer 3: In the Abstract the sentence containing the expression: " ... to maximize the percent released of AT for 12 h ..." needs correction.
  • Response to the reviewer:

The sentence has been corrected.

  1. Comment 2 from Reviewer 3: Same with the sentence containing: " ... tablets were, respectively found to obey the ...".
  • Response to reviewer:

The sentence has been corrected

  1. Comment 3 from Reviewer 3: The animal handling licence number is missing.
  • Response to reviewer:

Ethical committee approval and license number added.

  1. Comment 4 from Reviewer 3: In Introduction: Please check the sentence ending with: " ... gastrointestinal complications[10-12]." (a gap required before citations).
  • Response to reviewer:

A gap has been made.

  1. Comment 5 from Reviewer 3: Same with the sentence: "... The objective of the present work to formulate ...".
  • Response to the reviewer:

The sentence has been corrected

  1. Comment 6 from Reviewer 3: The main difference(s) of the present work with prior literature need to be clearly mentioned in the Introduction and Conclusion.
  • Response to reviewer:

The differences have been added.

  1. Comment 7 from Reviewer 3: Please correct the following sentence (in Introduction):  "The formulated tablets were evaluated as illustrated in a Figure 2). ".
  • Response to the reviewer:

The sentence has been corrected.

  1. Comment 8 from Reviewer 3: Also, in the above sentence a bracket is missing and the number 2 should be changed to 1 (it refers to Figure 1 not Figure 2).
  • Response to reviewer:

The sentence has been modified.

  1. Comment 9 from Reviewer 3: The quality of Figures 3 and 7 need to be improved.
  • Response to reviewer:

 We improved figures quality

  1. Comment 10 from Reviewer 3: In my opinion Figure legends need to be relatively self-explanatory and abbreviations such as AT  and  ROS-SDs need to be explained.
  • Response to reviewer:

The abbreviations have been replaced.

  1. Comment 11 from Reviewer 3: Several typing mistakes are present in Conclusion. For instance: ... management is typed "mangmant"! ... double "of" is used in the last sentence ... "... rabbits ... which was treated ...", ...
  • Response to the reviewer:

The words have been modified.

Round 2

Reviewer 2 Report

Thank you for your revised manuscript:

The following revisions need to be done to improve the publication:

in general:

The format is not optimized (units are separated from numbers and the interlining is heterogeneous. Please revise thoroughly!

Some minor spelling errors do persist (sections 2.4, 2.12, 2.14 just to name some sections).

The equations could be outlined (in italic or eq.).

Please take the time to re-examine the methods section for missing details on apparatuses (model, brand, city, country).

section 2.3: please add the information of your filter to the text (pores size & the membrane material, which is of outmost importance with poorly soluble drugs due to absorption phenomena).

section 2.11.2: The model of the Radwag balance and the city of the Radwag headquarters are missing in text.

section 2.11.3: The model of Mitutoyo micrometer is missing in text.

section 2.11.4: The model and brand of friability & hardness tester are missing in text.

section 2.11.6.1: The model and brand of USP type I dissolution tester are missing in text. What was the membrane material of the 0.45µm filter?

section 2.12: To avoid any misunderstanding could you rephrase, please “Secondly, ROS (immediate-release layer) was fed into the same cavity and compressed at 6 – 8 kg/cm² until the overall desired hardness of XXX unit was obtained.”

section 2.13: Details on humidity chamber are missing.

section 2.14.1: “The ZU-IACUC Committee of the Veterinary Faculty, Zagazig University, Egypt, has given its approval to this experiment's methods for the care of experimental rats. Application number ZU-IACUC/3/Az/89/2021. “ I am not an expert, but is the approval limited to rats? If not could you rephrase, because there might be further readers who might pay no credibility in the study if it is stated in the current from.

section 3.1.1: Please add the standard deviations to table 4 (can be calculated).

section 3.1.2: DSC analysis should be performed in triplicate, please add the missing data (triplicate data can be added as supplementary data).

section 3.1.3: For a better understanding, please indicate with arrows in Figure 3 the peaks that indicate non-interaction.

section 3.1.4: The X-Ray diffraction spectrum of the solid dispersion (Figure 4) shows a peak at around 2Theta= 13 degrees. How do you explain this peak? Is there some information in the literature? Additionally, are there any metastable forms of ROS reported?

section 3.2: Could you please add a more generous explanation on the fact of the unexpected order of angle of repose of the different formulations? As you replied: “that the difference in lactose amount is very small to cause a significant effect on flowability.” What does support your theory or is there a particular explanation (particle shape)? (My remark aims to have a close look on the data and to analyze it thoroughly, to gain further insight into the impact of formulation on powder properties).

Concerning my comment 19 (report 1):

    “Comment 19 from Reviewer 2: 3.7. The rosuvastatin release seems to slightly decrease during storage, do you have any information on the tablet’s stability in storage in non-sealed containers? Is there some information on this type of solid dispersion in the literature?”

    you replied: “Response to reviewer:

-We believe that there was no significant change in the release of rosuvastatin, this slight decrease in drug release may be due to a handling error or the drug content varies slightly from tablet to tablet.”

Next to your beliefs, can you furnish any scientific evidence (test repeated after further storage time?)

Thank you for your collaboration to increase the quality of this interesting paper.

Author Response

Dear Pharmaceutics Journal Editors: We're grateful that you shared the views of these respected reviewers with us. Here, we explain the effort that we put into revising our manuscript and responding to the reviewer's comments.

Reviewer 2 (2nd report)

Our team appreciates the opportunity to address your concerns and answer your questions in the following paragraphs:

In general:

Q1. The format is not optimized (units are separated from numbers and the interlining is heterogeneous. Please revise thoroughly!

  • Thank you very much, DONE.

Q2. Some minor spelling errors do persist (sections 2.4, 2.12, and 2.14 just to name some sections).

  • Revised as directed.

Q3. The equations could be outlined (in italic or eq.).

  •  

Q4. Please take the time to re-examine the methods section for missing details on apparatuses (model, brand, city, country).

  •  

Q5. Section 2.3: please add the information of your filter to the text (pores size & the membrane material, which is of utmost importance with poorly soluble drugs due to absorption phenomena).

  •  

Q6. Section 2.11.2: The model of the Radwag balance and the city of the Radwag headquarters are missing in the text.

  • Thanks, the data has been added, line 240.

Q7. Section 2.11.3: The model of the Mitutoyo micrometer is missing in the text.

  • The missing data has been added.

Q8. Section 2.11.4: The model and brand of friability & hardness tester are missing in the text.

  • The missing data has been added.

Q9. Section 2.11.6.1: The model and brand of the USP type I dissolution tester is missing in the text. What was the membrane material of the 0.45µm filter?

  • USP type I dissolution tester is VDS, Hanson Research Co., Massachusetts; USA.
  • The membrane material is a Whatman 0.45μm membrane filter.

Q10. Section 2.12: To avoid any misunderstanding could you rephrase, please “Secondly, ROS (immediate-release layer) was fed into the same cavity and compressed at 6 – 8 kg/cm² until the overall desired hardness of XXX unit was obtained.”

  •  

Q11. Section 2.13: Details on humidity chamber are missing.

  • DONE

Q12. Section 2.14.1: “The ZU-IACUC Committee of the Veterinary Faculty, Zagazig University, Egypt, has given its approval to this experiment's methods for the care of experimental rats. Application number ZU-IACUC/3/Az/89/2021. “I am not an expert, but is the approval limited to rats? If not could you rephrase, because there might be further readers who might pay no credibility to the study if it is stated in the current form?

  • NO, for all the experimental animals, accordingly it was revised in the text.

Q13. Section 3.1.1: Please add the standard deviations to table 4 (can be calculated).

  • DONE

Q14. Section 3.1.2: DSC analysis should be performed in triplicate, please add the missing data (triplicate data can be added as supplementary data).

  • We understand and agree with that point of view, and will consider it in our future work. But in this case similar results regarding ROS and Sorbitol obtained by Dhoranwala, et al, 2015, and Dezena, R.M.B. and Junior, J
  • Dhoranwala, K.A.; Shah, P.; Shah, S. Formulation optimization of rosuvastatin calcium-loaded solid lipid nanoparticles by 32 Full-factorial design. NanoWorld Journal.2015, 1.
  • Dezena, R.M.B.; Junior, J. Preformulation comparative study between two samples of sorbitol used as excipient in the direct compression process. J. Anal. Chem.2017, 4, 19-26.
  • Additionally the FT-IR and XRD confirm our results

Q15. Section 3.1.3: For a better understanding, please indicate with arrows in Figure 3 the peaks that indicate non-interaction.

  •  

Q16. Section 3.1.4: The X-Ray diffraction spectrum of the solid dispersion (Figure 4) shows a peak at around 2Theta= 13 degrees. How do you explain this peak? Is there some information in the literature? Additionally, are there any metastable forms of ROS reported?

  • There are no metastable forms of ROS reported, to our knowledge.
  • the test was redone on a new prepared sample and the new spectrum was added.

Q17. Section 3.2: Could you please add a more generous explanation on the fact of the unexpected order of angle of repose of the different formulations? As you replied: “that the difference in lactose amount is very small to cause a significant effect on flowability.” What does support your theory or is there a particular explanation (particle shape)? (My remark aims to have a close look at the data and to analyze it thoroughly, to gain further insight into the impact of formulation on powder properties).

Thanks for this valuable comment, so, we add the following discussion to section 3.2:

The angle of repose in ROS blend ranged from 16.59 to 29.23°, Although all the used excipients have good flowing properties the variation in the angle of repose may attribute to the concentration of superdisintegrant, increasing the concentration of superdisintegrant leading to a decrease in the angle of repose [1] as shown in table (5). Also, the angle of repose in AT blend ranged from 19.25 to 28.7°, Although all the used excipients have good flowing properties the variation of the obtained result may be due to the hygroscopic nature of the used cellulosic derivatives (HPMC and EC) [2] which may adsorb moisture from the atmosphere on their surface and form a cohesion force between the wetted particles and decrease flow properties [3]. The variation is obtained since the test of the different formulations proceeds on different days and the climatic condition is varied so the obtained result is varied but still in the accepted flowability range    

  1. Elkhodairy, K.A.; Hassan, M.A.; Afifi, S.A. Formulation and optimization of orodispersible tablets of flutamide. Saudi Pharmaceutical Journal.2014, 22, 53-61.
  2. Chambin, O.; Champion, D.; Debray, C.; Rochat-Gonthier, M.; Le Meste, M.; Pourcelot, Y. Effects of different cellulose derivatives on drug release mechanism studied at a preformulation stage. Journal of controlled release.2004, 95, 101-108.
  3. Fitzpatrick, J.; Barry, K.; Cerqueira, P.; Iqbal, T.; O’neill, J.; Roos, Y. Effect of composition and storage conditions on the flowability of dairy powders. International Dairy Journal.2007, 17, 383-392.

Concerning my comment 19 (report 1):

“Comment 19 from Reviewer 2: 3.7. The rosuvastatin release seems to slightly decrease during storage, do you have any information on the tablet’s stability in storage in non-sealed containers? Is there some information on this type of solid dispersion in the literature?” you replied: “Response to the reviewer:

-We believe that there was no significant change in the release of rosuvastatin, this slight decrease in drug release may be due to a handling error or the drug content varies slightly from tablet to tablet.”

Next to your beliefs, can you furnish any scientific evidence (test repeated after further storage time?)

  • After we refereed the literature, ROS has an acceptable decrease in its stability in some media, namely alkaline pH, photolytic sources, and an oxidative condition, in the range of (96% - 98%)
  • Source: Mukthinuthalapati MA, Bukkapatnam V, Bandaru SP. Stability indicates the liquid chromatographic method for the simultaneous determination of rosuvastatin and ezetimibe in pharmaceutical formulations. Adv Pharm Bull. 2014 Dec;4(4):405-11. DOI: 10.5681/apb.2014.060. Epub 2014 Aug 10. PMID: 25436199; PMCID: PMC4137433.

Round 3

Reviewer 2 Report

Thank you for the improvements.